# Ruminal fermentation, microbial population and lipid metabolism in gastrointestinal nematode-infected lambs fed a diet supplemented with herbal mixtures

**Paulina Szulc[1], Dominika Mravčáková[2], Malgorzata Szumacher-Strabel[1], Zora Váradyová[2], Marián Várady[3], Klaudia Čobanová[2], Linggawastu Syahrulawal[1], Amlan Kumar Patra[4], Adam Cieslak[1]***

**1** Department of Animal Nutrition, Poznan University of Life Sciences, Poznan, Poland, **2** Institute of Animal Physiology, Centre of Biosciences of Slovak Academy of Sciences, Košice, Slovak Republic, **3** Institute of Parasitology, Slovak Academy of Sciences, Košice, Slovak Republic, **4** Department of Animal Nutrition, West Bengal University of Animal and Fishery Sciences, Kolkata, India

* adam.cieslak@up.poznan.pl

**Data Availability Statement:** All relevant data are within the paper and its Supporting Information files.

## Abstract

The aim of this study was to evaluate the effects of medicinal herbal mixtures rich in phenolic, flavonoid and alkaloid compounds on ruminal fermentation and microbial populations, and fatty acid (FA) concentrations and lipid oxidation in tissues of lambs infected with the gastrointestinal nematode (GIN) parasite (*Haemonchus contortus*). Parallel *in vitro* and *in vivo* studies were performed using two different herbal mixtures (Mix1 and Mix2). The *in vitro* study was conducted in a 2 (infection status; non-infected versus infected) × 3 (diets; control, Mix1 and Mix2) factorial design. In the *in vivo* study, 24 lambs were equally divided into four treatments: non-infected lambs fed a control diet, infected lambs fed the control diet, infected lambs fed a diet with Mix1 and infected lambs fed a diet with Mix2. Herbal mixtures (100 g dry matter (DM)/d) were added to the basal diets of meadow hay (*ad libitum*) and a commercial concentrate (500 g DM/d). The experimental period lasted for 70 days. Ruminal fermentation characteristics and methane production were not affected by infection *in vivo* or *in vitro*. Both herbal mixture supplementation increased total volatile fatty acid (VFA) concentrations ($P < 0.01$) and DM digestibility ($P < 0.01$) *in vitro*. *Archaea* population was slightly diminished by both herbal mixtures ($P < 0.05$), but they did not lower methane production *in vitro* or *in vivo* ($P > 0.05$). Infection of *H. contortus* or herbal mixtures modulated FA proportion mainly in the liver, especially the long chain FA proportion. Concentrations of thiobarbituric acid reactive substances (TBARS) in serum were significantly higher after 70 days post-infection in the infected lambs. Herbal Mix1 supplementation reduced TBARS concentrations in meat after seven days of storage. In conclusion, supplementing of herbal mixtures to the diets of GIN parasite infected lambs did not affect the basic ruminal fermentation parameters. Herbal mixtures may improve few FA proportions mainly in liver as well as decrease lipid oxidation in meat.

**Funding:** This study was supported by funds from the Slovak Research and Development Agency (APVV 18-0131, APVV 17-0297) and by the framework of the Ministry of Science and Higher Education, Poland, programme "Regional Initiative Excellence" in years 2019-2022, Project No. 005/RID/2018/19. PSz is a PhD scholarship holder of the grant 2016/23/B/NZ9/03427 funded by National Science Center, Poland. LS has been awarded a full master degree by Ignacy Lukasiewicz scholarship from the Polish National Agency for Academic Exchange (NAWA). AC acknowledges the SAIA, n. o. (Slovak Academy Information Agency) for Academy Mobility Scholarship.

**Competing interests:** The authors have declared that no competing interests exist.

**Abbreviations:** ADF, acid detergent fiber; AU, arbitrary unit; CLA, conjugated linoleic acid; CI, control infected; CN, control non-infected; CP, crude protein; DI, desaturation index; DM, dry matter; ELOVL5, fatty acid elongase 5 (elongase 5); FA, fatty acids; FADS1, fatty acid desaturase 1 (Δ5-desaturase); FAME, fatty acids methyl ester; FASN, fatty acid synthase; GIN, gastrointestinal nematode; IVDMD, *in vitro* dry matter digestibility; LA, linoleic acid; LCFA, long chain fatty acids; LPL, lipoprotein lipase; M1I, Mix1 infected; M2I, Mix2 infected; MCFA, medium chain fatty acids; MDA, malondialdehyde; Mix1, infected; Mix1, herbal mixture 1; Mix1N, Mix 1 non-infected; Mix2, herbal mixture 2; Mix2I, Mix 2 infected; Mix2N, Mix2 non-infected; MUFA, monounsaturated fatty acids; NDF, neutral detergent fiber; PCR, polymerase chain reaction; PSM, plant secondary metabolites; PUFA, polyunsaturated fatty acids; RA, rumenic acid; RNA, ribonucleic acid; RT, reverse transcription; SCD, stearoyl-CoA desaturase (Δ9-desaturase); SFA, saturated fatty acids; TBARS, thiobarbituric acid reactive substances; UFA, unsaturated fatty acids; VA, vaccenic acid; VFA, volatile fatty acids.

## Introduction

Gastrointestinal parasitic infections is one of the major issues impacting the health of livestock animals, especially by the most pathogenic gastrointestinal nematode (GIN) parasite *Haemonchus contortus*. This GIN sucks abomasum blood and causes anemia, reduces reproductive capacity and animal production, resulting in considerable economic losses [1,2]. Since GIN reduces productivity, infected animals require more resource input to achieve the same level of productive output compared to the non-infected animals. Ovine periparturient parasitism increases greenhouse gas intensity; and therefore gastrointestinal parasite control could improve production efficiency and decrease environmental footprints in sheep production systems [3]. Chemoprophylaxis against *H. contortus* by application of anthelmintics repeatedly poses the risk of development of anthelmintic resistance and residues in food products [4]. Therefore, there is a growing interest in feeding of diets supplemented with plant secondary metabolites (PSM) to GIN infected animals for reducing the transmission of the parasites and the diseases associated with parasites [5,6]. The use of PSM has been beneficial to treat various digestive or parasitic disorders due to their nutraceutical and anthelmintic activities. Many studies favored natural sources of PSM such as *Hypericum perforatum*, *Malva parviflora*, *Prunella vulgaris*, *Juniperus communis*, *Pinus ponderosa*, *Melissa officinalis* and *Nepeta caesarea* as well as mixed medicinal herbs to reduce the burdens of GIN [7,8]. In the earlier studies, PSM that contains phytochemical substances such as flavonoids considers as important bioactive compound as antioxidant and antimicrobial properties in the rumen [9,10]. Another bioactive compound is polyphenol known as highly abundant groups of substances found in plants that can be classified based on a simple structure, for instance, phenolic acids and more complex such as tannins [11]. Polyphenols inhibit the populations and/or activity of microbes responsible for methanogenesis and biohydrogenation by among others changing the rumen environment (pH value) and through the toxic effect on methanogens, consequently lowering methane emission and biohydrogenation rate of UFA in the rumen [12,13,14]. The degree of ruminal fatty acid (FA) saturation affects FA composition in ruminant products such as meat and milk [15,16].

The lambs used in the present study were a part of a comprehensive experiment that investigated the effects of two dry mixtures of medicinal herbs on parasitological, inflammatory, antioxidant, and fecal microbiota composition in lambs experimentally infected with *H. contortus* [17]. In the present study, we hypothesized that the dietary dry medicinal herb mixtures may affect the ruminal methane production, FA concentrations in the liver, blood, subcutaneous fat and *musculus longissimus dorsi* muscle, lipid peroxidation and oxidative stability in meat due to their inhibitory effects on the ruminal methanogens and biohydrogenating microbial population and antioxidant properties. Infections with GIN in animals causes extra endogenous protein loss and increased energy metabolism, which subsequently may alter lipid metabolism and antioxidant status [18]. The influences of GIN on FA profile have not yet been studied in GIN-infected lambs. Therefore, our objective was to assess the supplementation of two medicinal herbal mixtures on ruminal fermentation characteristics, microbial population, methane production and lipid metabolism in GIN-infected lambs.

## Material and methods

Animals used and experimental design were approved by the Ethics Committee of the Institute of Parasitology of the Slovak Academy of Sciences, in accordance with European Community guidelines (EU Directive 2010/63/EU for animal experiments). Permission to collect samples and carry out the experiment was granted by the participating sheep farmers. Twenty-four Valachian female lambs with an initial mean body weight of 11.7 ± 1.23 kg and 3–4 months of

age lambs were obtained from the same farm. All animals were humanely killed at the end of the experiment (abattoir of the Centre of Biosciences of SAS, Institute of Animal Physiology, Košice, Slovakia, No. SK U 06018). The carcasses of animals were sent to the Department of Pathological Anatomy and Pathological Physiology, University of Veterinary Medicine and Pharmacy in Košice in Slovak Republic.

## Diet and supplements

This study was a part of a larger study that investigated natural chemotherapeutic alternatives for controlling of haemonchosis in lambs and had been described in more detail previously [17]. Animals were fed a concentrate mixture (500 g dry matter (DM)/d), herbal mixtures (non-commercial mixtures—Mix1 and Mix2; 100 g DM/d) and meadow hay (*ad libitum*). The concentrate mixture was composed of 700 g/kg of barley, 220 g/kg of soybean meal, 48 g/kg of wheat bran, 5 g/kg of bicarbonate and 27 g/kg of mineral-vitamin premix.

## Experimental design

***In vitro* experiment.**   The *in vitro* study was carried out using a batch culture system according to the modified protocol described previously [19]. Two herbal mixtures (Mix1 and Mix2) were used with 9 different herbs in each mixture. Dry herbs were obtained from commercial sources (AGROKARPATY, Plavnica, Slovak Republic and BYLINY Mikeš s.r.o., Číčenice, Czech Republic). Herbal composition of Mix1: stems of *Artemisia absinthium* L. (1%), *Fumaria officinalis* L. (13.4%), *Hyssopus officinalis* L. (13.4%), *Melissa officinalis* L. (13.4%) and *Solidago virgaurea* L. (13.4%); flowers of *Matricaria chamomilla* L. (13.4%) and *Malva sylvestris* L. (13.4%); leaves of *Plantago lanceolata* L. (13.4%) and seeds of *Foeniculum vulgare* Mill. (5%). The phytochemical substances of Mix1 contained 57.3 g/kg DM of phenolic acids and 41.5 g/kg DM of flavonoids with greater concentrations of myricetin 3-O-galactoside (20.2 g/kg DM), 1,5-dicaffeoylquinic acid (15.4 g/kg DM),3-O-caffeoylquinic acid (11.3 g/kg DM), and dihydrocaffeoyl-4-caffeoyl quinic acid (9.72 g/kg DM) [17]. Herbal composition of Mix2: stems of *Artemisia absinthium* L. (1%), *Malva sylvestris* L. (12.4%), *Achillea milefolium* L. (12.4%), *Cichorium intybus* L. (12.4%), *Hypericum perforatum* L. (12.4%) and *Urtica dioica* L. (12.4%); flowers of *Matricaria chamomilla* L. (12.4%), *Fumaria officinalis* L. (12.4%) and *Calendula officinalis* L. (12.4%). The phytochemical substances of Mix2 contained 22.2 g/kg DM of phenolic acids and 29.5 g/kg DM of flavonoids with high concentrations of 3-O-caffeoylquinic acid (6.91 g/kg DM), 1,5-Dicaffeoylquinic acid (6.18 g/kg DM), rutin (5.73 g/kg DM) and 2-O-feruloylhydroxycitric acid (3.64 g/kg DM).Protoberberine-type alkaloids were also present in Mix1 (1.4 g/kg DM) and Mix2 (1.33 g/kg DM) [17].

For the *in vitro* study, the ruminal content was collected from the top, bottom and middle of the rumen of each lamb separately. The fresh ruminal content was collected at a slaughter house from six control non-infected (CN) and six control infected (CI) lambs with two CN and two CI at each run. Infection status was identified at autopsy by observing the *H. contortus* worms after the opening of the abomasum. The *in vitro* study was completed in three runs and total 12 lambs were used. The same diet was used as a control in the *in vivo* trial. After slaughtering of lambs, rumen digesta was taken from different parts (top, bottom and middle) of the rumen. The experiment was conducted in a 2 infection status (non-infection and infection) × 3 diets (control, Mix1 and Mix2) factorial arrangement with following 6 treatments: Control diet with non-infection (CN), Mix1 diet with non-infection (Mix1N), and Mix2 diet with non-infection (Mix2N), Control diet with infection (CI), Mix1 diet with infection (Mix1I), and Mix2 diet with infection (Mix2I). Ruminal content was squeezed through a four-layer cheese-cloth into two separate Schott Duran® bottles (SCHOTT North America, Inc. Corporate

Office, Elmsford, NY 10523, USA) and immediately transported to the laboratory in a 39 ˚C preheated water bath. Two bottles were used for collecting rumen fluid separately for CN and CI. Five replicate bottles in each treatment (6 treatments × 5 bottles) were used in three consecutive runs. The ruminal fluid was diluted with buffer solution at a ratio of 1:4, and buffered fluid was transferred to the bottles with prepared substrates anaerobically. The control groups (CN and CI) contained 400 mg of substrate (252 mg DM of hay and 148 mg DM of the commercial concentrate). For the herbal mixture, 36 mg DM (9% of 400 mg substrate) of Mix1 or Mix2 was further added to the 400 mg substrate. The bottles with buffered ruminal fluid and substrate were filled with $CO_2$, closed with rubber stoppers and sealed with aluminum cups. Then the bottles were incubated in an incubator (Galaxy 170R, Eppendorf North America Inc., Hauppauge, NY) for 24 h at a temperature of 39 ˚C in an anaerobic condition with periodical mixing of the contents.

***In vivo* experiment.**    Based on the *in vitro* results, the *in vivo* experiment was designed. Twenty-four Improved Valachian female lambs with an initial mean body weight of 11.7 ± 1.23 kg and 3–4 months of age were kept in stalls for 15 d for adaptation to the diet. During the whole experiment, lambs had free access to drinking tap water. After the adaptive period, the lambs were divided into four treatment groups (n = 6): non-infected control group (CN), GIN-infected group fed with the control diet (CI), infected group fed the control diet supplemented with Mix1 (M1I) or Mix2 (M2I). Lambs were infected orally with 5000 third-stage larvae of the MHco1 (strain of *H. contortus*), which is susceptible to all main classes of anthelmintics. Infection increased egg counts in the infected animals as shown previously [17]. Lambs were fed with a basal diet of meadow hay *ad libitum* and a commercial concentrate at 500 g DM/day in the control groups for the growth rate of 150 g/d (Table 1). Commercial concentrate was composed of 700 g/kg of barley, 220 g/kg of soybean meal, 48 g/kg of wheat bran, 5 g/kg of bicarbonate and 27 g/kg of mineral-vitamin premix. In the herbal mixture groups, Mix1 and Mix2 were additionally fed at 100 g dry matter (DM)/day to the M1I and M2I lambs, respectively. The experimental period was 70 days (during summer), and the animals were housed on a sheep farm.

## Sample analysis

**Chemical composition of feed.**    Chemical composition of dietary ingredients was analyzed in triplicates by standard procedures [20]. The dry matter (DM) content was determined by drying the samples at 105 ˚C for 48 h in a hot air oven. The ash content was determined by burning the samples at 550 ˚C for 12 h (method no. 942.05) in a muffle furnace (Nabertherm, LT 40/12, GmbH, Lilienthal, Germany. Nitrogen (N) content (method no. 968.06) was determined using a FLASH 400 Analyzer (Thermo Fisher Scientific, Cambridge, UK). Crude protein (CP) content was calculated by multiplying the N content by 6.25 (method no. 990.03). The acid-detergent fiber (ADF) and neutral detergent fiber (NDF) contents were determined as described previously [21] by using a FiberCap system (FiberCap ™ 2021/2023, FOSS Analytical AB, Höganäs, Sweden). In forages (i.e., meadow hay, Mix1 and Mix2), NDF was assayed without a heat-stable amylase and expressed inclusive of residual ash. In concentrate, NDF was assayed with a heat-stable amylase and expressed inclusive of residual ash. ADF was expressed inclusive of residual ash.

**Basic ruminal fermentation.**    After 24 h of in vitro incubation, the volume of accumulated gas released from the batch culture was determined from the recorded pressure or the volume of gas produced after 24 h of fermentation using a mechanical manometer fitted to a transducer (Premagas, Stará Turá, Slovak Republic). Analysis of gas production was carried out by gas chromatography using a PerkinElmer Clarus 500 gas chromatograph (Perkin Elmer, Inc.,

**Table 1. Chemical composition and fatty acid profile of the diets.**

| Item | Meadow hay | Concentrate | Mix1 | Mix2 |
|---|---|---|---|---|
| **Main chemical composition, g/kg DM** | | | | |
| **CP** | 163 | 309 | 160 | 180 |
| **aNDF** | 825 | 140 | 500 | 460 |
| **ADF** | 500 | 90 | 360 | 350 |
| **Ash** | 39 | 29 | 110 | 110 |
| **Fatty acid proportion, g/100 g of FA** | | | | |
| **C12:0** | 1.06 | 0.11 | 0.12 | 0.41 |
| **C14:0** | 0.90 | 0.34 | 0.36 | 1.57 |
| **C16:0** | 18.6 | 14.0 | 12.5 | 25.0 |
| **C18:0** | 5.09 | 2.26 | 3.22 | 8.84 |
| **C18:1 *cis*-9** | 14.5 | 19.4 | 22.3 | 8.8 |
| **C18:2 *cis*-9 *cis*-12** | 36.3 | 55.6 | 26.9 | 25.3 |
| **C18:3 *cis*-9 *cis*-12 *cis*-15 (ALA)[a]** | 9.50 | 2.46 | 11.9 | 9.28 |
| **C20:3n-6** | 1.95 | 0.23 | 1.04 | 0.78 |
| **C20:5n-3 (EPA)[b]** | 0.19 | 0.05 | 0.19 | 0.09 |
| **C22:5n-3 (DPA)[c]** | 0.35 | 0.06 | 0.22 | 0.42 |
| **C22:6n-3 (DHA)[d]** | 1.21 | 0.20 | 0.30 | 0.42 |
| **Other FA[e]** | 10.3 | 5.29 | 20.9 | 19.1 |
| **SFA[f]** | 29.4 | 18.0 | 17.9 | 37.8 |
| **UFA[g]** | 70.6 | 82.0 | 82.1 | 62.2 |
| **MUFA[h]** | 20.7 | 22.9 | 41.1 | 26.0 |
| **PUFA[i]** | 49.8 | 59.2 | 41.0 | 36.2 |
| **n-6** | 38.6 | 56.4 | 28.4 | 26.3 |
| **n-3** | 25.9 | 22.2 | 12.6 | 9.84 |

[a] ALA, [α]-Linolenic acid.

[b] EPA, Eicosapentaenoic acid.

[c] DPA, Docosapentaenoic acid.

[d] DHA, Docosahexaenoic acid.

[e] Other FA, (C10:0, C14:1, C15:1, C16:1, C18:1 c11, C20:0, C22:1 n-9, C22:0, C23:0, C24:1)

[f] SFA, Saturated fatty acids.

[g] UFA, Unsaturated fatty acids.

[h] MUFA, Monounsaturated fatty acids.

[i] PUFA, Polyunsaturated fatty acids.

Shelton, CT, USA). The ruminal fluid was then collected from each bottle for analysis of pH, volatile fatty acids (VFA) and ammonia concentrations, and ruminal microorganism populations (bacteria, protozoa, and methanogens). For the *in vivo* experiment, ruminal fluid samples were collected immediately after slaughtering the animals. The pH value was measured immediately after sample collection using a pH meter (CP-104; Elmetron, Zabrze, Poland). Methane concentration from *in vitro* samples was determined by gas chromatography on PerkinElmer Clarus 500 gas chromatograph (Perkin Elmer, Inc., Shelton, USA) as described previously [22]. In the *in vivo* study, methane production was calculated measuring the molar proportion of VFA in the rumen as follow: 57.5 mol glucose = 65 mol acetate + 20 mol propionate + 15 mol butyrate + 60 mol $CO_2$ + 35 mol $CH_4$ + 25 mol $H_2O$. [23]. The concentration of ammonia-N was determined in the inocula by the phenol-hypochlorite method [24]. The VFA samples were analyzed by gas chromatography (PerkinElmer Clarus 500 gas chromatograph,

Perkin Elmer, Inc., Shelton, USA) as described previously [22]. The *in vitro* DM digestibility (IVDMD) and volume of accumulated gas were determined as described previously [22].

**Rumen microbial quantification.**   The total protozoa count in collected ruminal fluid was determined according to the previous method [25]. For bacterial quantification, DNA from the ruminal samples were isolated using a Mini Bead-Beater (BioSpec, Bartlesville, OK, USA) for cell lysis, followed by purification (QIAamp DNA Stool Mini Kit; Qiagen, Hilden, Germany) [26]. DNA concentrations and quality were measured with NanoDrop 2000 spectrophotometer (Thermo Scientific, Wilmington, DE, USA). The primers for the targeted species were *Butyrivibrio proteoclasticus* (F: *CCTAGTGTAGCGGTGAAATG*", R: *TTAGCGACGGCA CTGAATGCCTA*) [27], *Butyrivibrio fibrisolvens* (F: *ACACACCGCCCGTCACA*, R: *TCCTTACGGTTGGGTCACAGA*) [28], *Ruminococucus flavefaciens* (F: *CGAACGGAGATAA TTTGAGTTTACTTAGG*, R: *CGGTCTCTGTATGTTATGAGGTATTACC*) [29], *Fibrobacter succinogenes*, (F: *GTTCGGAATTACTGGGCGTAAA*, R: *CGCCTGCCCCTGAACTATC*) [29], and *Ruminococcus albus* (F: *CCCTAAAAGCAGTCTTAGTTCG*, R: *CCTCCTTGCGGTTAG AACA*) [30] for the quantitative PCR method. For the total bacteria, the following primers were used (F: *GTGATGCATGGTTGTCGTCA*, R: *GAGGAAGGTGKGGATGACGT*) [31].

Methanogens and total bacteria were quantified by the fluorescence *in situ* hybridization technique [32]. The rumen fluid (50 μl) was diluted in phosphate-buffered saline and pipetted onto 0.22 μm polycarbonate filters (Frisentte K02BP02500) and vacuumed (Vaccum KNF Vacuport-Neuberg). The filters were transferred onto a cellulose disk for dehydration in an ethanol concentration at different level (500, 800, and 900 ml/L) for 3 min. Hybridization was carried out in 50 μl of hybridization buffer (0.9 M NaCl; 20 mM Tris/HCl, pH 7.2; 0.1 g/L of SDS) containing oligonucleotide probes (all methanogens (S-D-Arch-0915-a-A-20) and two order-specific probes: S-O-Mmic-1200-a-A-21) (*Methanomicrobiales*) and S-F-Mbac-0310-a-A-22 (*Methanobacteriales*) [33]. The filters were washed with washing buffer (20 mM Tris/ HCl, pH 7.2; 0.1 g/L of SDS; 5 mM EDTA) for 20 minutes at 48 ˚C. The filters were then rinsed gently in distilled water, air-dried and mounted on object glasses with VectaShield (Vector laboratories nr. H-1000) anti-fading agent containing DAPI (4',6-diamidino-2-phenylindole). To distinguish the total count of bacteria (DAPI) from other methanogens in the rumen fluid, filters were maintained at 4 ˚C for 1 h in the dark until visualization using an Axio Imager M2 microscope (Carl Zeiss Iberia, Madrid, Spain).

**Fatty acids extraction and analysis.**   On the last day of experiment, the lambs were slaughtered and samples from *longissimus dorsi* muscle, subcutaneous fat and liver were collected. The muscle samples (approximately 200 g) were collected from the right side of each carcass and drawn at the level of 13[th] thoracic rib. Samples of subcutaneous fat and liver were lyophilized by freezing, vacuuming and drying the samples (Epsilon 2-10D LSCplus, CHRIST, Germany). Samples of muscle were lyophilized after removing the epimysium. All collected samples were stored at -80 ˚C until lipid extraction [34]. The FA concentrations in feeds, liver, muscle, and subcutaneous fat [15], ruminal fluid [13] and blood [35] were determined using standard protocols [15]. FA were identified and quantified based on peaks and retention times by comparing FA sample target with appropriate fatty acids methyl ester (FAME) standards (37 FAME Mix, Sigma-Aldrich) and the concentrations of CLAs were determined using a CLA standard (a mixture of cis 9, trans 11 and trans 10, cis 12-octadecadienoic acid methyl esters; Sigma-Aldrich) using a Galaxie Work Station 10.1 (Varian, CA).

**Gene expression with RT-qPCR.**   Samples of *longissimus dorsi* muscle were collected immediately after slaughter and shock frozen in liquid nitrogen. Relative transcript abundances of five lipogenic genes such as lipoprotein lipase (LPL), fatty acid synthase (FASN), stearoyl-CoA desaturase (SCD), fatty acid desaturase 1 (FADS1), fatty acid elongase 5 (ELOVL5) were measured by real-time PCR method as described previously [10]. The

muscle samples were homogenized in 1 ml TriPure reagent (Roche Diagnostics, Mannheim, Germany) using Tissue Lyser II (Qiagen, USA). Then the RNA isolation was performed following the protocol provided by the manufacturer. Briefly, 200 μl of chloroform (Sigma Aldrich, Hamburg, Germany) was added into tubes and shaken. After 10 min, samples were centrifuged (15 min) at 12,000 g speed. The clear phase was transferred to a new tube and added with 0.5 ml isopropanol (Sigma Aldrich, Hamburg, Germany). Then probes were centrifuged (15 min) at 12 000 g speed once again. RNA pellets were washed with 750 ml/L of ethanol (POCH, Gliwice, Poland), centrifuged for the third time (10 min at 9000 g) and dried at 40 ˚C thermoblock (Eppendorf, Hamburg, Germany). The RNA was then resuspended in DEPC treated water (Invitrogen, Carlsbad, USA) for spectrophotometric measurement (Nanodrop c2000, Thermo Scientific, USA) of concentration and purity. A reverse transcription reaction (RT) was performed using a Transcriptor First Strand cDNA Synthesis Kit (Roche) according to the procedures described by the manufacturer. Each sample was adjusted to equal concentrations of RNA. Briefly, RNA (300 ng), random hexameters (60 μM), oligodT (2.5 mM) and water were mixed and denatured at 65 ˚C for 10 min. Reverse transcriptase and RNase inhibitor buffer were then added to the RNA mix to a final volume of 20 μl. The RT conditions were as follows: 25 ˚C for 5 min, followed by 42 ˚C for 45 min and 85 ˚C for 5 min. The gene expression of FA synthase (FASN), lipoprotein lipase (LPL), stearoyl-CoA desaturase (SCD), FA desaturase 1 (FADS1) and FA elongase 5 (ELOVL5) were measured in muscle. Primer pairs for RT-qPCR amplification were designed based on previously published oligonucleotides [36] and synthesized by Sigma-Aldrich (USA). Only standard curves with an efficiency of at least 1.9 were considered optimized for the reaction in particular conditions. RT-qPCR amplification was performed in duplicate on a Light Cycler 480 instrument (Roche Diagnostics, Germany) using Light Cycler Sybr Green 480 I Master (Bio-Rad, USA). The RT-qPCR mix (10 μl per sample) contained 2 μl of nuclease-free water, 2 μl of primers mix, 5 μl Sybr Green Master mix and 1 μl of cDNA. The RT-qPCR conditions were as follows: 95 ˚C, 5 min (pre-incubation); 40 cycles of: 95 ˚C, 5 s (denaturation); 60 ˚C, 12 s (primer annealing and elongation); 65–97 ˚C (PCR product melting). For each RT-qPCR run, a negative control sample (without cDNA) was also added. After each analysis, melting curves were checked to exclude any potential sample contamination. Relative gene expression was evaluated by delta delta CT (ΔΔCT) with Gapdh/beta actin as a reference.

**Blood analyses.** Blood samples were collected from the jugular vein of each animal on day 22, 37, 51 and 70 into 10-ml serum-separator tubes (Sarstedt AG & Co, Nümbrecht, Germany) and centrifuged at 1200 g for 10 min at room temperature. From all collected days, the serum samples were used for lipid peroxidation. For FA analysis, sera from day 70 were used. The sera were stored at—80 ˚C until analysis.

**Lipid oxidation.** The left *m. longissimus dorsi* muscle samples were excised within 15 min after the slaughter and were immediately vaccum packed. Meat oxidative stability was monitored in the muscle samples that were stored at 4 ˚C for 0, 1 or 7 days. The standard curve of malondialdehyde prepared by hydrolysis of 1,1,3,3,-tetraethoxypropane (Sigma-Aldrich) was used to assess the lipid oxidation by the thiobarbituric acid reactive substances (TBARS) method as described previously [37].

## Calculations

The desaturase [38], atherogenic [39] and thrombogenic [21] indices were calculated from the FA profile. Methane and hydrogen production, and hydrogen utilization were estimated based on stoichiometry calculations [23].

**Statistical analysis.**    All data were analyzed using SAS statistical software (Univ. Edition, version 9.4) [40]. In experiment 1 (*in vitro* study), data were analyzed using PROC MIXED procedure with models containing treatment group, infection, and their interaction as fixed factors and each consecutive run was considered as a random factor. In experiment 2 (*in vivo* study), data except for the lipid peroxidation were analyzed with one way ANOVA model with PROC GLM procedure. Two-way ANOVA (GraphPad Prism, GraphPad Software, Inc., San Diego, USA) was used for the analysis of lipid oxidation in serum and meat to test the effect of dietary treatment and the time of sampling/storage, as well as their interaction. The significant differences among treatment groups were tested with Tukey post-hoc test ($P < 0.05$). All values are shown as the means with pooled standard errors of means.

## Results

### *In vitro* experiment

The pH decreased due to infection ($P < 0.01$), but Mix2N group had also decreased the pH compared to the CN ($P = 0.01$; Table 2). The IVDMD of Mix1N, Mix2N, Mix1I, and Mix2I was improved compared to either the non-infected or infected control ($P < 0.01$). The gas produced in CI decreased compared to CN ($P = 0.03$), but was similar in the infected and non-infected groups supplemented with Mix1 and Mix2 ($P = 0.02$). Mix1N group produced more methane compared to CN and Mix1I ($P < 0.02$). However, methane production in Mix1I was lower than the Mix1N and Mix2I when $CH_4$ was expressed as $CH_4$/gas produced and $CH_4$/IVDMD ($P = 0.03$ and $P = 0.05$, respectively). Concentrations of total VFA were lower in CN group compared with the groups supplemented with Mix1 and Mix2 ($P < 0.01$). The acetic acid proportion decreased in all infected groups compared to the non-infected control ($P < 0.01$), but the iso-valerate and valerate concentrations in all infected groups increased compared to the CN.

Regarding the ruminal microbial activity, the *Archaea* populations of Mix1 and Mix2 in both non-infected and infected animals were lower compared to the CN ($P < 0.01$). Total bacterial abundance in the Mix2N group was lower compared to all groups ($P < 0.05$). The relative abundance of *R. albus* tended to increase in CI compared to the CN ($P < 0.08$). Also, *F. succinogenes* abundance was higher in infected groups ($P < 0.01$) and significantly lower in Mix1N and Mix2N. The relative abundance of *B. proteoclasticus* was higher in the Mix1I compared to the CI or CN and also to other groups. In contrast, the *B. fibrisolvens* of the Mix1I was lower than in CI and CN and also than other groups ($P < 0.01$).

Regarding the FA concentration in the buffered rumen fluid, major changes occurred due to the infection for C16:0, C18:0, C18:1 *trans*-10, C18:1 *trans*-11, C18:2 *cis*-9 *cis*-12; docosapentaenoic acid (DPA), docosahexaenoic acid (DHA), saturated FA (SFA), UFA, PUFA, n6 FA, n6/n3 ratio, medium chain FA (MCFA), and long chain FA (LCFA) (Table 3). The lower proportions of α-linolenic acid (ALA) were found in rumen fluid treated with Mix2N, Mix1I and Mix2I compared to the CN and the CI. Herbal mixtures changed the FA concentration in the ruminal fluid. The C18:1 *trans*-11 and the SFA proportions of all herbal groups with infected and non-infected were higher compared to CN ($P < 0.01$); whereas higher UFA proportions of CN were noted compared to other groups except for the Mix2N ($P = 0.02$).

### *In vivo* experiment

There were no significant differences ($P > 0.05$) among the groups for ruminal fermentation characteristics in lambs (Table 4). The bacteria population (*B. fibrisolvens*, *R. albus* and *F. succinogenes*) of the infected lambs fed with control diet as well as infected lambs treated with Mix1 and Mix2 diets increased ($P < 0.01$); however other bacterial populations did not differ among the treatment groups except *B. proteoclasticus*, which had higher relative abundance in

**Table 2. The effect of herbal mixtures and infection on the rumen fermentation and microbial populations *in vitro*.**

| Parameter[a] | Non-infected | | | Infected[b] | | | SEM | P | | |
|---|---|---|---|---|---|---|---|---|---|---|
| | CN | Mix1N | Mix2N | CI | Mix1I | Mix2I | | I | G | I×G |
| **pH** | 6.24[a] | 6.21[a] | 6.08[b] | 6.12[b] | 6.04[b] | 6.04[b] | 0.01 | <0.01 | <0.01 | 0.01 |
| **IVDMD, %** | 53.9[c] | 62.9[a] | 62.2[ab] | 54.1[c] | 63.3[a] | 60.0[b] | 0.70 | 0.47 | <0.01 | 0.29 |
| **$NH_3$, mM** | 6.06[b] | 7.32[a] | 6.63[ab] | 6.09[ab] | 6.44[ab] | 6.10[ab] | 0.13 | 0.08 | 0.03 | 0.29 |
| **Gas produced, ml** | 66.3[a] | 67.9[a] | 67.2[a] | 59.1[b] | 66.9[a] | 66.9[a] | 0.94 | 0.03 | <0.01 | 0.02 |
| **$CH_4$, mM** | 0.57[b] | 0.80[a] | 0.59[ab] | 0.61[ab] | 0.46[b] | 0.71[ab] | 0.04 | 0.37 | 0.70 | 0.02 |
| **$CH_4$/Gas produced, mM/ml** | 0.008[ab] | 0.011[a] | 0.009[ab] | 0.010[ab] | 0.007[b] | 0.011[a] | 0.001 | 0.74 | 0.67 | 0.03 |
| **$CH_4$/IVDMD, mM/g** | 2.51[ab] | 3.21[a] | 2.74[ab] | 2.83[ab] | 1.92[b] | 3.13[a] | 0.20 | 0.52 | 0.63 | 0.05 |
| **Total VFA, mM** | 52.5[c] | 57.1[a] | 55.7[ab] | 53.7[bc] | 56.0[ab] | 57.9[a] | 0.42 | 0.23 | <0.01 | 0.17 |
| Acetate, mol/100 mol | 63.8[a] | 63.1[ab] | 62.9[ab] | 61.6[b] | 60.8[b] | 61.2[b] | 0.30 | <0.01 | 0.32 | 0.88 |
| Propionate, mol/100 mol | 20.4 | 20.4 | 20.4 | 21.3 | 22.0 | 21.5 | 0.29 | 0.06 | 0.87 | 0.88 |
| Isobutyrate, mol/100 mol | 0.28[b] | 0.35[ab] | 0.30[ab] | 0.30[b] | 0.36[a] | 0.31[ab] | 0.01 | 0.51 | <0.01 | 1.00 |
| Butyrate, mol/100 mol | 13.1 | 13.6 | 13.9 | 13.2 | 13.0 | 13.3 | 0.11 | 0.09 | 0.24 | 0.36 |
| Isovalerate, mol/100 mol | 0.79[b] | 0.86[ab] | 0.83[ab] | 0.88[a] | 0.93[a] | 0.92[a] | 0.01 | <0.01 | <0.01 | 0.81 |
| Valerate, mol/100 mol | 1.31[b] | 1.41[b] | 1.40[b] | 2.45[a] | 2.56[a] | 2.50[a] | 0.1 | <0.01 | 0.78 | 0.99 |
| Caproate, mol/100 mol | 0.25 | 0.29 | 0.28 | 0.36 | 0.38 | 0.37 | 0.02 | <0.01 | 0.64 | 0.95 |
| **A:P** | 3.2 [a] | 3.14 [a] | 3.11 [a] | 3.01 [b] | 2.85 [b] | 2.94 [b] | 0.05 | 0.05 | 0.64 | 0.90 |
| *Archaea*, $10^7$/ml | 1.07[a] | 0.88[b] | 0.56[c] | 0.86[b] | 0.61[c] | 0.54[c] | 0.05 | 0.05 | <0.01 | 0.49 |
| Total bacteria, $10^8$/ml | 4.94[a] | 4.82[a] | 4.06[b] | 5.56[a] | 5.28[a] | 5.58[a] | 0.15 | <0.01 | 0.47 | 0.31 |
| *R. albus*, AU[c] | 1.29[b] | 1.0[b] | 0.25[b] | 11.63[a] | 0.64[b] | bd | 1.43 | 0.08 | 0.12 | 0.07 |
| *R. flavefaciens*, AU | 0.09 | Bd | 0.03 | 0.03 | bd | bd | 0.03 | 0.48 | 0.43 | ND |
| *F. succinogenes*, AU | 0.58[c] | 0.50[c] | 0.19[c] | 2.95[a] | 1.7[b] | 1.38[b] | 0.29 | <0.01 | 0.16 | 0.42 |
| *B. proteoclasticus*, AU | 0.79[c] | 0.05[c] | 0.13[c] | 2.93[b] | 8.87[a] | 4.26[b] | 0.66 | <0.01 | <0.01 | <0.01 |
| *B. fibrisolvens*, AU | 2.26[b] | 4.07[ab] | 0.51[bcd] | 1.19[c] | 0.19[d] | 4.54[a] | 0.43 | 0.61 | 0.54 | <0.01 |
| Total protozoa, $10^3$/ml | 67.0 | 66.9 | 68.8 | 71.0 | 68.3 | 74.8 | 0.03 | 0.11 | 0.40 | 0.78 |
| Holotricha, $10^3$/ml | 0.71 | 0.55 | 0.59 | 0.51 | 0.45 | 0.61 | 1.15 | 0.10 | 0.24 | 0.23 |
| Entodiniomorpha, $10^3$/ml | 66.3 | 66.3 | 68.2 | 70.5 | 67.9 | 74.2 | 1.15 | 0.10 | 0.41 | 0.78 |

Within each row, means with lower case superscripts (a–d) indicate significant differences at $P < 0.05$; SEM, standard error of the mean.

[a]IVDMD, *in vitro* dry mater digestibility; VFA, volatile fatty acids; bd, below detection.

[b]Control non-infected (CN); Mix 1 non-infected (Mix1N); Mix2 non-infected (Mix2N); Control infected (CI); Mix1 infected (Mix1I); Mix 2 infected (Mix2I); I, infected; G, group.

[c]AU, The relative 16S rRNA gene copy abundance expressed as an arbitrary unit relative the total bacterial gene copy abundance of the control.

the infected M2I group ($P < 0.01$). The population of *Holotricha* was higher in the CI than other groups ($P < 0.01$).

The FA proportions in the ruminal fluid, blood, as well as in the liver, subcutaneous fat and *m. longissimus dorsi* varied. The proportions of C15:0 and C17:0 in the rumen were higher in M2I lambs compared with the CI lambs whereas the proportions of C14:1 and C17:1 in the rumen were higher in M2I lambs compared with the CI and CN lambs ($P < 0.05$; Table 5). The ruminal MCFA proportion of CI was lower than the M2I ($P = 0.03$). By contrast, ruminal LCFA proportion was higher in the CI lambs than in the M2I lambs ($P = 0.03$).

In the serum from lambs fed Mix2, C15:0, C16:0, C16:1, C18:1 *trans*-6-8, ALA, C18:2 *trans*-10 *cis*-12, and MCFA proportions were higher compared to the CI group (Table 6). The M2I had lower proportions of C18:1 *cis*-11 and C18:2 *cis*-9 *cis*-12 in serum compared to other groups, which led to the lowered PUFA and LCFA proportions. The serum from lambs fed Mix1 and Mix2 had the lowest n6/n3 FA ratio compared to the CI ($P < 0.001$).

**Table 3. The effect of herbal mixtures and infection on ruminal fatty acid proportions (g/100 g FA)** *in vitro*.

| Fatty acids, g/100 g FA | Non-infected | | | Infected[a] | | | SEM | P value | | |
|---|---|---|---|---|---|---|---|---|---|---|
| | CN | Mix1N | Mix2N | CI | Mix1I | Mix2I | | I | G | I×G |
| **Saturated** | | | | | | | | | | |
| C8:0 | 0.11[ab] | 0.14[a] | 0.11[ab] | 0.08[b] | 0.12[ab] | 0.10[ab] | 0.01 | 0.14 | 0.04 | 0.83 |
| C10:0 | 0.07[ab] | 0.08[a] | 0.06[ab] | 0.05[b] | 0.06[ab] | 0.04[ab] | 0.004 | 0.01 | 0.13 | 0.98 |
| C12:0 | 1.03[ab] | 1.18[a] | 0.89[b] | 1.05[ab] | 1.05[ab] | 0.84[b] | 0.03 | 0.42 | 0.02 | 0.57 |
| C13:0 | 8.66[ab] | 9.72[a] | 8.91[ab] | 7.82[b] | 7.89[ab] | 7.47[b] | 0.20 | <0.01 | 0.37 | 0.50 |
| C14:0 | 1.59 | 1.79 | 1.74 | 1.77 | 1.90 | 1.89 | 0.04 | 0.06 | 0.09 | 0.90 |
| C15:0 | 1.39 | 1.54 | 1.53 | 1.46 | 1.42 | 1.34 | 0.03 | 0.15 | 0.69 | 0.11 |
| C16:0 | 22.7[a] | 23.5[a] | 23.1[a] | 21.2[b] | 21.1[b] | 20.8[b] | 0.18 | <0.01 | 0.50 | 0.36 |
| C17:0 | 0.96[ab] | 1.06[a] | 1.09[a] | 0.88[b] | 0.89[b] | 0.94[ab] | 0.02 | <0.01 | 0.02 | 0.35 |
| C18:0 | 27.1[b] | 27.2[b] | 28.6[b] | 32.5[a] | 32.1[a] | 33.4[a] | 0.40 | <0.01 | 0.05 | 0.75 |
| **Monounsaturated** | | | | | | | | | | |
| C14:1 | 0.58 | 0.71 | 0.68 | 0.71 | 0.74 | 0.69 | 0.02 | 0.12 | 0.11 | 0.23 |
| C15:1 | 1.04 | 1.20 | 1.08 | 1.10 | 1.14 | 1.16 | 0.02 | 0.53 | 0.17 | 0.44 |
| C16:1 | 0.61[a] | 0.42[b] | 0.43[b] | 0.47[b] | 0.42[b] | 0.38[b] | 0.02 | 0.04 | <0.01 | 0.13 |
| C17:1 | 0.23 | 0.19 | 0.17 | 0.22 | 0.21 | 0.27 | 0.01 | 0.06 | 0.61 | 0.08 |
| C18:1 *trans*-6-8 | 0.47 | 0.51 | 0.53 | 0.47 | 0.45 | 0.56 | 0.01 | 0.64 | 0.02 | 0.42 |
| C18:1 *trans*-9 | 0.45[b] | 0.46[b] | 0.51[ab] | 0.63[a] | 0.54[ab] | 0.67[a] | 0.02 | <0.01 | 0.20 | 0.51 |
| C18:1 *trans*-10 | 0.76[c] | 1.00[bc] | 1.09[bc] | 1.36[ab] | 1.41[ab] | 1.70[a] | 0.08 | <0.01 | 0.04 | 0.72 |
| C18:1 *trans*-11 | 2.80[c] | 4.04[a] | 4.11[a] | 3.20[b] | 3.87[ab] | 3.70[ab] | 0.09 | 0.70 | <0.01 | 0.04 |
| C18:1 *cis*-9 | 11.2[a] | 8.84[b] | 8.81[bc] | 8.52[bc] | 8.26[bc] | 8.08[c] | 0.23 | <0.01 | <0.01 | 0.05 |
| C18:1 *cis*-11 | 1.24[c] | 1.33[abc] | 1.41[abc] | 1.32[bc] | 1.55[a] | 1.44[ab] | 0.02 | 0.01 | <0.01 | 0.17 |
| C18:1 *cis*-12 | 0.23[b] | 0.30[ab] | 0.35[a] | 0.29[ab] | 0.37[a] | 0.35[a] | 0.01 | 0.01 | <0.01 | 0.18 |
| C18:1 *cis*-13 | 0.18 | 0.18 | 0.21 | 0.16 | 0.19 | 0.14 | 0.01 | 0.03 | 0.74 | 0.10 |
| C18:1 *cis*-14 | 0.37[b] | 0.39[ab] | 0.43[ab] | 0.45[a] | 0.47[a] | 0.46[a] | 0.01 | <0.01 | 0.19 | 0.52 |
| **Polyunsaturated** | | | | | | | | | | |
| C18:2 *cis*-9 *cis*-12 | 8.21[a] | 6.64[b] | 7.05[ab] | 6.84[b] | 6.66[b] | 6.18[b] | 0.17 | 0.03 | 0.02 | 0.21 |
| C18:3 *cis*-9 *cis*-12 *cis*-15 (ALA)[b] | 0.50[a] | 0.45[a] | 0.15[b] | 0.47[a] | 0.16[b] | 0.13[b] | 0.04 | 0.04 | <0.01 | 0.08 |
| C18:2 *cis*-9 *trans*-11 (RA/CLA)[c] | 0.94 | 0.84 | 1.10 | 0.81 | 1.00 | 0.94 | 0.04 | 0.50 | 0.16 | 0.11 |
| C18:2 *trans*-10 *cis*-12 | 0.23 | 0.21 | 0.22 | 0.24 | 0.25 | 0.22 | 0.01 | 0.20 | 0.59 | 0.44 |
| C18:3n6 | 0.17 | 0.17 | 0.16 | 0.14 | 0.13 | 0.16 | 0.01 | 0.02 | 0.76 | 0.43 |
| C20:2 | 0.06 | 0.04 | 0.05 | 0.06 | 0.05 | 0.05 | 0.01 | 0.63 | 0.57 | 0.94 |
| C20:3n6 | 1.24[a] | 1.09[ab] | 0.87[b] | 0.88[b] | 0.99[ab] | 1.04[ab] | 0.04 | 0.21 | 0.47 | 0.01 |
| C20:4n6 | 0.06 | 0.05 | 0.07 | 0.06 | 0.04 | 0.05 | 0.003 | 0.13 | 0.15 | 0.23 |
| C20:5n3 (EPA)[d] | 0.16 | 0.12 | 0.11 | 0.10 | 0.11 | 0.16 | 0.01 | 0.63 | 0.75 | 0.04 |
| C22:2 | 0.06 | 0.05 | 0.06 | 0.05 | 0.05 | 0.03 | 0.003 | 0.02 | 0.62 | 0.28 |
| C22:5n3 (DPA)[e] | 0.42[a] | 0.21[ab] | 0.23[ab] | 0.20[b] | 0.29[ab] | 0.33[ab] | 0.03 | 0.88 | 0.59 | 0.01 |
| C22:6n3 (DHA)[f] | 1.68[b] | 1.91[ab] | 2.00[ab] | 2.05[a] | 1.84[ab] | 2.00[ab] | 0.04 | 0.25 | 0.42 | 0.04 |
| SFA[g] | 63.9[b] | 67.3[a] | 67.0[a] | 67.8[a] | 67.4[a] | 67.4[a] | 0.35 | 0.02 | 0.05 | 0.01 |
| UFA[h] | 35.6[a] | 32.4[b] | 32.8[ab] | 31.9[b] | 32.2[b] | 31.9[b] | 0.34 | 0.01 | 0.05 | 0.02 |
| MUFA[i] | 21.4 | 20.6 | 20.8 | 20.0 | 20.7 | 20.6 | 0.22 | 0.26 | 1.00 | 0.28 |
| PUFA[j] | 13.7[a] | 11.8[b] | 12.0[ab] | 11.9[b] | 11.5[b] | 11.3[b] | 0.21 | 0.02 | 0.01 | 0.19 |
| n6 FA | 10.0[a] | 8.3[b] | 8.5[ab] | 8.3[b] | 8.2[ab] | 7.8[b] | 0.19 | 0.02 | 0.04 | 0.16 |
| n3 FA | 2.67 | 2.67 | 2.46 | 2.82 | 2.45 | 2.62 | 0.05 | 0.76 | 0.10 | 0.24 |
| n6/n3 ratio | 3.77[a] | 2.95[ab] | 3.56[ab] | 3.03[b] | 3.47[ab] | 3.14[ab] | 0.10 | 0.26 | 0.69 | 0.02 |
| MCFA[k] | 37.7[b] | 40.3[a] | 38.5[b] | 35.8[c] | 35.5[bc] | 34.3[c] | 0.32 | <0.01 | 0.08 | 0.04 |

*(Continued)*

**Table 3.** (Continued)

| Fatty acids, g/100 g FA | Non-infected | | | Infected[a] | | | SEM | P value | | |
|---|---|---|---|---|---|---|---|---|---|---|
| | CN | Mix1N | Mix2N | CI | Mix1I | Mix2I | | I | G | I×G |
| LCFA[l] | 61.7[bc] | 59.4[c] | 61.3[bc] | 64.0[a] | 64.1[ab] | 65.0[a] | 0.32 | <0.01 | 0.11 | 0.11 |

Within each row, means with lower case superscripts (a–c) indicate significant differences at $P < 0.05$; SEM, standard error of the mean.

[a] Control non-infected (CN); Mix 1 non-infected (Mix1N); Mix2 non-infected (Mix2N); Control infected (CI); Mix1 infected (Mix1I); Mix 2 infected (Mix2I); I, infected; G, group.

[b] ALA, [α]-Linolenic acid.

[c] RA/CLA, Rumenic acid/Conjugated linoleic acid.

[d] EPA, Eicosapentaenoic acid.

[e] DPA, Docosapentaenoic acid.

[f] DHA, Docosahexaenoic acid.

[g] SFA, Saturated fatty acids.

[h] UFA, Unsaturated fatty acids.

[i] MUFA, Monounsaturated fatty acids.

[j] PUFA, Polyunsaturated fatty acids.

[k] MCFA, Medium chain fatty acids.

[l] LCFA, Long chain fatty acids.

In the liver of animals fed both herbal mixtures, proportions of C16:0, C16:1, and MCFA, and DI (16:1/16) decreased compared to CN and CI (Table 7). However, the increased ALA, n3 FA, LCFA proportions ($P < 0.01$) in M1I and M2I compared to the CI group were observed.

Among the various FA profiles in the *longissimus dorsi* muscle, significant ($P < 0.03$) changes in C16:0 in M2I and C20:5 n-3 in M1I compared to CN were noticed. The MCFA significantly decreased ($P < 0.01$) compared to CN and CI and LCFA significantly increased ($P < 0.02$) in the M1I and M2I compared to CI (Table 8).

The subcutaneous fat from M2I group was characterized by higher proportions of C15:0, C14:1, C18:1 *cis*-12, and C18:1 *cis*-14 compared to the CN and CI (Table 9). The M1I group had higher proportions of C18:0 compared only to the CI ($P < 0.05$). Both herbal mixture groups had higher proportions of C18:1 *cis*-14 and α-linolenic acid (ALA) in the subcutaneous fat. The M2I group had decreased MUFA proportion and CI (MUFA/SFA), and M1I group had decreased n6/n3 ratio compared to the CI group.

The CN and M2I groups had lower relative transcript abundances of LPL compared with the CI group ($P = 0.01$) (Table 10). Lower relative transcript abundances of FASN in the CN lambs compared to the M2I lambs ($P = 0.03$) and lower relative transcript abundances of SCD in the CN lambs compared with the M1I lambs ($P = 0.04$) were observed. Also, lower relative transcript abundances of FADS1 in the M2I group compared to the M1I group ($P < 0.01$) were detected. The gene expression of ELOVL5 was not changed in any group.

The TBARS level in serum was influenced by time ($P < 0.001$), with significantly higher values after 70 days post-infection in the CI lambs compared with the CN lambs (Table 11). The TBARS levels in the meat were also affected by the time of storage ($P < 0.001$) and by the groups, which was higher in the CI group compared to CN and M1I groups ($P < 0.05$).

## Discussion

It is well known that gastrointestinal endoparasites increase metabolic and nutritional demand of the host, which is manifested by impaired growth, productivity, reproductive ability and

**Table 4. The effect of herbal mixtures on rumen fermentation and microbial populations in lambs with *H. contortus* infection.**

| Item | CN[a] | CI[a] | M1I[a] | M2I[a] | SEM | *P* value |
|---|---|---|---|---|---|---|
| **pH** | 6.75 | 6.49 | 6.62 | 6.85 | 0.06 | 0.17 |
| **NH$_3$, mM** | 9.44 | 8.40 | 8.81 | 8.79 | 0.22 | 0.39 |
| **CH$_4$, mM** | 0.39 | 0.40 | 0.42 | 0.40 | 0.01 | 0.96 |
| **CH$_4$ production, mM** | 19.2 | 19.6 | 21.2 | 19.5 | 0.69 | 0.73 |
| **H$_2$ production, mM** | 126 | 137 | 137 | 127 | 4.12 | 0.71 |
| **H$_2$ utilization, mM** | 114 | 123 | 123 | 114 | 3.71 | 0.71 |
| **Total VFA, mM** | 63.8 | 70.2 | 68.7 | 64.6 | 2.06 | 0.68 |
| Acetate, mol/100 mol | 68.8 | 64.2 | 69.8 | 69.2 | 0.80 | 0.06 |
| Propionate, mol/100 mol | 18.1 | 20.8 | 17.0 | 17.5 | 0.82 | 0.43 |
| Isobutyrate, mol/100 mol | 0.44 | 0.40 | 0.30 | 0.43 | 0.05 | 0.81 |
| Butyrate, mol/100 mol | 10.5 | 11.3 | 10.7 | 10.4 | 0.38 | 0.88 |
| Isovalerate, mol/100 mol | 0.80 | 0.79 | 0.47 | 0.69 | 0.08 | 0.47 |
| Valerate, mol/100 mol | 1.26 | 2.31 | 1.69 | 1.61 | 0.14 | 0.08 |
| Caproate, mol/100 mol | 0.14 | 0.28 | 0.17 | 0.18 | 0.03 | 0.30 |
| **A:P ratio** | 3.97 | 3.40 | 4.16 | 4.13 | 0.19 | 0.53 |
| ***Archaea*, 10$^7$/ ml** | 1.03 | 0.96 | 0.70 | 0.94 | 0.07 | 0.96 |
| **Total bacteria, 10$^8$/ml** | 4.65[b] | 5.95[a] | 6.06[a] | 5.98[a] | 0.20 | <0.01 |
| *B. fibrisolvens*, AU [b] | 0.03 | 0.02 | 0.01 | 0.06 | 0.01 | 0.08 |
| *B. proteoclasticus*, AU | 0.06[b] | 0.08[b] | 0.04[b] | 0.57[a] | 0.07 | <0.01 |
| *R. albus*, AU | 0.02 | 0.03 | 0.05 | 0.05 | 0.01 | 0.10 |
| *F. succinogenes*, AU | 0.20 | 0.40 | 0.45 | 0.33 | 0.07 | 0.64 |
| **Total protozoa, 10$^4$/ml** | 45.7 | 40.2 | 66.5 | 71.0 | 5.00 | 0.07 |
| Entodiniomorpha, 10$^4$/ml | 45.3 | 39.7 | 66.2 | 70.6 | 5.01 | 0.07 |
| Holotricha, 10$^4$/ml | 0.34[b] | 0.51[a] | 0.29[b] | 0.32[b] | 0.02 | <0.01 |

Within each row, means with lower case superscripts (a,b) indicate significant differences at *P* < 0.05; SEM, standard error of the mean.

[a]Control non-infected (CN); Control infected (CI); Mix1 infected (M1I); Mix2 infected (M2I).

[b]AU, The relative 16S rRNA gene copy abundance expressed as an arbitrary unit relative the total bacterial gene copy abundance of the control.

**Table 5. The effect of herbal mixtures on fatty acid proportions in ruminal fluid (g/100 g FA) in lambs with *H. contortus* infection.**

| Fatty acids, g/100 g FA | CN[a] | CI[a] | M1I[a] | M2I[a] | SEM | *P* value |
|---|---|---|---|---|---|---|
| **Saturated** | | | | | | |
| **C8:0** | 0.05 | 0.04 | 0.04 | 0.05 | 0.00 | 0.64 |
| **C10:0** | 0.03 | 0.03 | 0.04 | 0.04 | 0.01 | 0.91 |
| **C12:0** | 0.65[a] | 0.43[b] | 0.42[b] | 0.49[ab] | 0.03 | 0.02 |
| **C13:0** | 3.36 | 2.89 | 4.64 | 6.05 | 0.63 | 0.18 |
| **C14:0** | 1.10 | 0.95 | 0.88 | 1.06 | 0.08 | 0.82 |
| **C15:0** | 1.62[ab] | 1.18[b] | 1.70[ab] | 2.19[a] | 0.13 | 0.02 |
| **C16:0** | 24.9 | 23.0 | 22.8 | 24.7 | 0.86 | 0.32 |
| **C17:0** | 0.78[ab] | 0.74[b] | 0.86[ab] | 0.93[a] | 0.03 | 0.04 |
| **C18:0** | 27.5 | 30.1 | 29.7 | 26.4 | 1.08 | 0.62 |
| **Monounsaturated** | | | | | | |
| **C14:1** | 0.94[b] | 0.72[b] | 1.03[b] | 1.39[a] | 0.07 | <0.01 |
| **C15:1** | 1.34 | 1.11 | 1.05 | 1.40 | 0.07 | 0.22 |
| **C16:1** | 0.45 | 0.33 | 0.34 | 0.40 | 0.02 | 0.20 |
| **C17:1** | 0.22[b] | 0.23[b] | 0.24[ab] | 0.31[a] | 0.01 | 0.02 |

(*Continued*)

**Table 5.** (Continued)

| Fatty acids, g/100 g FA | CN[a] | CI[a] | M1I[a] | M2I[a] | SEM | P value |
|---|---|---|---|---|---|---|
| C18:1 *trans*-6-8 | 0.32 | 0.55 | 0.45 | 0.25 | 0.08 | 0.57 |
| C18:1 *trans*-9 | 0.38 | 0.56 | 0.43 | 0.30 | 0.05 | 0.35 |
| C18:1 *trans*-10 | 0.66 | 0.71 | 0.83 | 0.49 | 0.68 | 0.19 |
| C18:1 *trans*-11 | 2.96 | 2.88 | 3.64 | 2.77 | 0.17 | 0.29 |
| C18:1 *cis*- 9 | 9.38 | 9.30 | 8.01 | 7.51 | 0.46 | 0.42 |
| C18:1 *cis*-11 | 1.05 | 1.30 | 1.04 | 0.93 | 0.08 | 0.36 |
| Polyunsaturated | | | | | | |
| C18:2 *cis*-9 *cis*-12 | 11.5 | 11.3 | 11.8 | 10.7 | 0.37 | 0.78 |
| C18:3 *cis*-9 *cis*-12 *cis*-15 (ALA)[a] | 0.52 | 0.10 | 0.28 | 1.44 | 0.21 | 0.46 |
| C18:2 *cis*-9 *trans*-11(RA/CLA)[b] | 1.84 | 3.41 | 1.73 | 1.96 | 0.59 | 0.72 |
| C18:2 *trans*-10 *cis*-12 | 0.20 | 0.24 | 0.19 | 0.18 | 0.01 | 0.36 |
| C18:3n6 | 0.10 | 0.08 | 0.06 | 0.11 | 0.01 | 0.32 |
| C20:2 | 0.21 | 0.07 | 0.01 | 0.14 | 0.04 | 0.42 |
| C20:3n6 | 0.66 | 0.68 | 0.83 | 1.14 | 0.08 | 0.15 |
| C20:4n6 | 0.08 | 0.09 | 0.06 | 0.08 | 0.01 | 0.58 |
| C20:5n3 (EPA)[d] | 0.08 | 0.07 | 0.06 | 0.03 | 0.01 | 0.30 |
| C22:2 | 0.07 | 0.05 | 0.08 | 0.05 | 0.00 | 0.16 |
| C22:5n3 (DPA)[e] | 0.19 | 0.17 | 0.22 | 0.15 | 0.03 | 0.84 |
| C22:6n3 (DHA)[f] | 2.51 | 2.13 | 2.19 | 2.81 | 0.13 | 0.21 |
| SFA[g] | 61.4 | 61.2 | 62.6 | 63.5 | 1.27 | 0.43 |
| UFA[h] | 38.6 | 38.8 | 37.4 | 36.5 | 1.27 | 0.43 |
| MUFA[i] | 20.3 | 20.0 | 19.6 | 17.7 | 0.99 | 0.24 |
| PUFA[l] | 18.3 | 18.8 | 17.8 | 18.8 | 0.66 | 0.96 |
| n6 FA | 12.7 | 12.6 | 13.3 | 12.4 | 0.36 | 0.86 |
| n3 FA | 3.61 | 2.90 | 3.06 | 4.43 | 0.23 | 0.07 |
| n6/n3 | 3.79 | 4.76 | 4.38 | 2.84 | 0.33 | 0.19 |
| MCFA[k] | 34.4[ab] | 28.1[b] | 32.8[ab] | 37.7[a] | 1.29 | 0.03 |
| LCFA[l] | 65.5[ab] | 71.8[a] | 67.1[ab] | 62.2[b] | 1.30 | 0.03 |

Within each row, means with lower case superscripts (a–c) indicate significant differences at $P < 0.05$; SEM, standard error of the mean.

[a] Control non-infected, CN; Control infected, CI; Mix1 infected, M1I; Mix2 infected, M2I.

[b] ALA, [α]-Linolenic acid.

[c] RA/CLA, Rumenic acid/Conjugated linoleic acid.

[d] EPA, Eicosapentaenoic acid.

[e] DPA, Docosapentaenoic acid.

[f] DHA, Docosahexaenoic acid.

[g] SFA, Saturated fatty acids.

[h] UFA, Unsaturated fatty acids.

[i] MUFA, Monounsaturated fatty acids.

[j] PUFA, Polyunsaturated fatty acids.

[k] MCFA, Medium chain fatty acids.

[l] LCFA, Long chain fatty acids.

reduction in feed intake up to 20–25% [41]. Limited research is available on the effect of the GIN infection affecting ruminal fermentation and lipid metabolism profile in small ruminants. Also, periparturient parasitism in sheep may increase greenhouse gas emission [3]. A recent study showed that parasite infections in lambs can increase in methane yield (g $CH_4$/kg) by

**Table 6. The effect of herbal mixture on fatty acid proportions (g/100 g FA) in the serum of lambs with *H. contortus* infection.**

| Fatty acids, g/100 g FA | CN[a] | CI[a] | M1I[a] | M2I[a] | SEM | *P* value |
|---|---|---|---|---|---|---|
| **Saturated** | | | | | | |
| C8:0 | 0.11 | 0.06 | 0.11 | 0.22 | 0.02 | 0.14 |
| C10:0 | 0.31 | 0.35 | 0.41 | 0.39 | 0.04 | 0.86 |
| C12:0 | 0.18 | 0.14 | 0.37 | 0.52 | 0.07 | 0.18 |
| C14:0 | 0.49 | 0.28 | 0.40 | 0.65 | 0.06 | 0.14 |
| C15:0 | 0.56[ab] | 0.45[b] | 0.69[ab] | 1.11[a] | 0.09 | 0.03 |
| C16:0 | 13.4[b] | 12.0[b] | 11.0[b] | 17.5[a] | 0.65 | <0.01 |
| C17:0 | 0.59 | 0.42 | 0.44 | 0.85 | 0.07 | 0.13 |
| C18:0 | 12.8 | 15.3 | 15.7 | 13.5 | 0.66 | 0.36 |
| **Monounsaturated** | | | | | | |
| C14:1 | 0.24 | 0.20 | 0.39 | 0.43 | 0.05 | 0.28 |
| C15:1 | 0.19 | 0.10 | 0.19 | 0.34 | 0.03 | 0.09 |
| C16:1 | 1.35[ab] | 0.97[b] | 0.39[b] | 1.68[a] | 0.13 | <0.01 |
| C17:1 | 0.52 | 0.49 | 0.45 | 0.58 | 0.04 | 0.71 |
| C18:1 *trans*- 6–8 | 0.16[b] | 0.10[b] | 0.20[b] | 0.47[a] | 0.04 | <0.01 |
| C18:1 *trans*- 9 | 0.07 | 0.09 | 0.10 | 0.16 | 0.01 | 0.13 |
| C18:1 *trans*- 10 | 0.18 | 0.35 | 0.24 | 0.46 | 0.06 | 0.45 |
| C18:1 *trans*- 11 | 0.48 | 0.74 | 0.91 | 0.92 | 0.08 | 0.13 |
| C18:1 *cis*-9 | 20.0[ab] | 17.6[ab] | 15.9[b] | 20.4[a] | 0.64 | 0.02 |
| C18:1 *cis*-11 | 3.26[a] | 3.49[a] | 2.49[a] | 1.17[b] | 0.24 | <0.01 |
| C18:1 *cis*-12 | 0.58 | 0.67 | 0.54 | 0.35 | 0.05 | 0.19 |
| C18:1 *cis*-13 | 0.13 | 0.16 | 0.13 | 0.12 | 0.02 | 0.95 |
| C18:1 *cis*-14 | 0.21 | 0.17 | 0.21 | 0.31 | 0.03 | 0.52 |
| **Polyunsaturated** | | | | | | |
| C18:2 *cis*-9 *cis*-12 | 31.1[a] | 34.3[a] | 33.1[a] | 22.5[b] | 1.29 | <0.01 |
| C18:3 *cis*-9 *cis*-12 *cis*-15 (ALA)[b] | 2.50[bc] | 1.95[c] | 3.14[ab] | 3.50[a] | 0.17 | <0.01 |
| C18:2 *cis*-9 *trans*-11(RA/CLA)[c] | 0.09 | 0.10 | 0.09 | 0.16 | 0.01 | 0.41 |
| C18:2 *trans*-10 *cis*-12 | 0.14[ab] | 0.11[b] | 0.20[ab] | 0.29[a] | 0.02 | 0.02 |
| C18:3n6 | 0.10 | 0.09 | 0.11 | 0.19 | 0.02 | 0.05 |
| C20:2 | 0.17 | 0.13 | 0.21 | 0.12 | 0.02 | 0.14 |
| C20:3n6 | 4.33 | 4.28 | 5.41 | 4.53 | 0.21 | 0.15 |
| C20:4n6 | 0.31 | 0.14 | 0.20 | 0.23 | 0.05 | 0.72 |
| C20:5n3 (EPA)[d] | 0.29 | 0.29 | 0.35 | 0.25 | 0.03 | 0.72 |
| C22:2 | 0.25 | 0.15 | 0.19 | 0.22 | 0.03 | 0.70 |
| C22:5n3 (DPA)[e] | 1.06 | 1.04 | 1.54 | 1.35 | 0.08 | 0.04 |
| C22:6n3 (DHA)[f] | 0.27 | 0.33 | 0.29 | 0.30 | 0.04 | 0.98 |
| SFA[g] | 30.3 | 30.4 | 30.6 | 36.6 | 1.04 | 0.07 |
| UFA[h] | 69.7 | 69.6 | 69.4 | 63.4 | 1.04 | 0.07 |
| MUFA[i] | 29.0[a] | 26.7[ab] | 24.5[b] | 29.8[a] | 0.68 | 0.01 |
| PUFA[j] | 40.7[ab] | 42.9[a] | 44.9[a] | 33.6[b] | 1.32 | <0.01 |
| n6 FA | 36.7[a] | 39.6[a] | 39.6[a] | 28.0[b] | 1.39 | <0.01 |
| n3 FA | 4.12[ab] | 3.61[b] | 5.32[a] | 5.40[a] | 0.23 | <0.01 |
| n6/n3 | 9.15[ab] | 11.2[a] | 7.59[bc] | 5.32[c] | 0.57 | <0.01 |
| MCFA[k] | 16.4[b] | 14.1[b] | 13.4[b] | 22.2[a] | 0.88 | <0.01 |
| LCFA[l] | 83.2[a] | 85.4[a] | 86.1[a] | 77.2[b] | 0.89 | <0.01 |
| **Desaturation index** | | | | | | |
| DI (16:1/16) | 0.09[a] | 0.07[a] | 0.03[b] | 0.09[a] | 0.01 | <0.01 |

*(Continued)*

**Table 6.** (Continued)

| Fatty acids, g/100 g FA | CN[a] | CI[a] | M1I[a] | M2I[a] | SEM | P value |
|---|---|---|---|---|---|---|
| DI (18:1/18) | 0.38 | 0.46 | 0.50 | 0.40 | 0.02 | 0.03 |
| DI (MUFA/SFA) | 0.49 | 0.47 | 0.44 | 0.45 | 0.01 | 0.25 |
| DI(20:4n6/20:3n6) | 0.06 | 0.04 | 0.04 | 0.05 | 0.01 | 0.74 |
| DI (20:4n6/18:3n6) | 0.63 | 0.59 | 0.62 | 0.55 | 0.04 | 0.90 |
| DI (22:6n3/22:5n3) | 0.19 | 0.23 | 0.15 | 0.18 | 0.02 | 0.66 |
| Thrombogenic index | 0.61 | 0.64 | 0.57 | 0.70 | 0.03 | 0.36 |
| Atherogenicity index | 0.42 | 0.41 | 0.41 | 0.54 | 0.02 | 0.12 |

Within each row, means with lower case superscripts (a–c) indicate significant differences at $P < 0.05$; SEM, standard error of the mean.

[a] Control non-infected, CN; Control infected, CI; Mix1 infected, M1I; Mix2 infected, M2I.

[b] ALA, [α]-Linolenic acid.

[c] RA/CLA, Rumenic acid/Conjugated linoleic acid.

[d] EPA, Eicosapentaenoic acid.

[e] DPA, Docosapentaenoic acid.

[f] DHA, Docosahexaenoic acid.

[g] SFA, Saturated fatty acids.

[h] UFA, Unsaturated fatty acids.

[i] MUFA, Monounsaturated fatty acids.

[j] PUFA, Polyunsaturated fatty acids.

[k] MCFA, Medium chain fatty acids.

[l] LCFA, Long chain fatty acids.

**Table 7. The effect of herbal mixture on fatty acid proportions (g/100 g FA) in the liver of lambs with *H. contortus* infection.**

| Fatty acids, g/100 g FA | CN[a] | CI[a] | M1I[a] | M2I[a] | SEM | P value |
|---|---|---|---|---|---|---|
| **Saturated** | | | | | | |
| C8:0 | 0.06 | 0.07 | 0.05 | 0.05 | 0.01 | 0.78 |
| C10:0 | 0.06 | 0.11 | 0.06 | 0.04 | 0.01 | 0.10 |
| C12:0 | 0.21 | 0.24 | 0.17 | 0.14 | 0.02 | 0.45 |
| C13:0 | 0.16 | 0.11 | 0.12 | 0.13 | 0.02 | 0.80 |
| C14:0 | 0.66[a] | 0.51[ab] | 0.33[b] | 0.42[ab] | 0.04 | 0.01 |
| C15:0 | 0.56 | 0.43 | 0.47 | 0.52 | 0.03 | 0.36 |
| C16:0 | 13.1[a] | 13.6[a] | 11.1[b] | 11.1[b] | 0.34 | <0.01 |
| C17:0 | 1.55 | 1.45 | 1.32 | 1.36 | 0.05 | 0.44 |
| C18:0 | 18.9[c] | 19.3[bc] | 21.7[ab] | 21.9[a] | 0.43 | 0.01 |
| **Monounsaturated** | | | | | | |
| C14:1 | 0.16 | 0.11 | 0.15 | 0.16 | 0.02 | 0.62 |
| C15:1 | 0.17 | 0.13 | 0.19 | 0.19 | 0.01 | 0.36 |
| C16:1 | 1.52[a] | 1.49[a] | 0.39[b] | 0.49[b] | 0.15 | <0.01 |
| C17:1 | 0.78[ab] | 0.82[a] | 0.47[b] | 0.51[ab] | 0.05 | 0.01 |
| C18:1 *trans*-6-8 | 0.24[b] | 0.39[a] | 0.27[ab] | 0.19[b] | 0.02 | 0.01 |
| C18:1 *trans*-9 | 0.29 | 0.35 | 0.24 | 0.24 | 0.02 | 0.10 |
| C18:1 *trans*-10 | 0.25 | 1.13 | 0.42 | 0.20 | 0.14 | 0.05 |
| C18:1 *trans*-11 | 0.61 | 0.70 | 1.05 | 0.85 | 0.08 | 0.21 |
| C18:1 *cis*-9 | 16.6 | 17.1 | 13.3 | 15.0 | 0.56 | 0.06 |
| C18:1 *cis*-11 | 1.45[ab] | 1.67[a] | 1.02[b] | 1.05[b] | 0.08 | <0.01 |
| C18:1 *cis*-12 | 0.14 | 0.14 | 0.17 | 0.18 | 0.02 | 0.82 |

(*Continued*)

**Table 7.** (Continued)

| Fatty acids, g/100 g FA | CN[a] | CI[a] | M1I[a] | M2I[a] | SEM | *P* value |
|---|---|---|---|---|---|---|
| **C18:1 *cis*-13** | 0.07 | 0.27 | 0.06 | 0.05 | 0.04 | 0.14 |
| **C18:1 *cis*-14** | 0.22 | 0.25 | 0.25 | 0.24 | 0.02 | 0.94 |
| **Polyunsaturated** | | | | | | |
| **C18:2 *cis*-9 *cis*-12** | 10.7 | 10.1 | 11.1 | 9.94 | 0.27 | 0.46 |
| **C18:3 *cis*-9 *cis*-12 *cis*-15 (ALA)[b]** | 1.04[ab] | 0.64[b] | 1.26[a] | 1.97[a] | 0.15 | 0.01 |
| **C18:2 *cis*-9 *trans*-11(RA/CLA)[c]** | 0.32 | 0.25 | 0.35 | 0.31 | 0.02 | 0.40 |
| **C18:2 *trans*-10 *cis*-12** | 0.26[a] | 0.23[ab] | 0.17[ab] | 0.14[b] | 0.02 | 0.03 |
| **C18:3n6** | 0.08 | 0.14 | 0.09 | 0.07 | 0.01 | 0.22 |
| **C20:2** | 1.88[a] | 1.79[ab] | 1.10[b] | 1.32[ab] | 0.11 | 0.03 |
| **C20:3n6** | 8.71 | 9.02 | 10.5 | 9.61 | 0.27 | 0.09 |
| **C20:4n6** | 0.50[a] | 0.33[ab] | 0.19[b] | 0.19[b] | 0.04 | 0.02 |
| **C20:5n3 (EPA)[d]** | 1.57 | 1.84 | 1.75 | 1.52 | 0.07 | 0.30 |
| **C22:2** | 0.39 | 0.39 | 0.22 | 0.25 | 0.03 | 0.06 |
| **C22:5n3 (DPA)[e]** | 4.65 | 4.28 | 6.10 | 5.42 | 0.30 | 0.14 |
| **C22:6n3 (DHA)[f]** | 0.14 | 0.13 | 0.15 | 0.18 | 0.01 | 0.27 |
| **SFA[g]** | 40.2 | 39.7 | 39.6 | 40.4 | 0.31 | 0.83 |
| **UFA[h]** | 59.8 | 60.3 | 60.4 | 59.6 | 0.31 | 0.83 |
| **MUFA[i]** | 29.6 | 31.1 | 27.4 | 28.7 | 0.53 | 0.08 |
| **PUFA[j]** | 30.2 | 29.2 | 32.9 | 30.9 | 0.61 | 0.17 |
| **n6 FA** | 20.5 | 20.1 | 22.2 | 20.2 | 0.43 | 0.28 |
| **n3 FA** | 7.39[ab] | 6.90[b] | 9.25[a] | 9.08[a] | 0.37 | 0.04 |
| **n6/n3** | 2.84[ab] | 3.03[a] | 2.40[ab] | 2.26[b] | 0.11 | 0.03 |
| **MCFA[k]** | 16.5[a] | 16.6[a] | 12.9[b] | 13.2[b] | 0.51 | <0.01 |
| **LCFA[l]** | 83.4[b] | 83.2[b] | 87.0[a] | 86.7[a] | 0.51 | <0.01 |
| **Desaturation index** | | | | | | |
| **DI (16:1/16)** | 0.10[a] | 0.10[a] | 0.03[b] | 0.04[b] | 0.01 | <0.01 |
| **DI (18:1/18)** | 0.54[b] | 0.53[b] | 0.62[a] | 0.59[ab] | 0.01 | 0.01 |
| **DI (MUFA/SFA)** | 0.42 | 0.44 | 0.41 | 0.42 | 0.01 | 0.13 |
| **DI (20:4n6/20:3n6)** | 0.64 | 0.50 | 0.65 | 0.73 | 0.03 | 0.09 |
| **DI (20:4n6/18:3n6)** | 0.86 | 0.68 | 0.70 | 0.72 | 0.03 | 0.23 |
| **DI (22:6n3/22:5n3)** | 0.03 | 0.03 | 0.02 | 0.04 | 0.00 | 0.51 |
| **Thrombogenic index** | 0.03 | 0.02 | 0.01 | 0.02 | 0.00 | 0.28 |
| **Atherogenicity index** | 0.29[a] | 0.27[ab] | 0.22[b] | 0.23[ab] | 0.01 | 0.02 |

Within each row, means with lower case superscripts (a–c) indicate significant differences at *P* < 0.05; SEM, standard error of the mean.

[a] Control non-infected, CN; Control infected, CI; Mix1 infected, M1I; Mix2 infected, M2I.

[b] ALA, [α]-Linolenic acid.

[c] RA/CLA, Rumenic acid/Conjugated linoleic acid.

[d] EPA, Eicosapentaenoic acid.

[e] DPA, Docosapentaenoic acid.

[f] DHA, Docosahexaenoic acid.

[g] SFA, Saturated fatty acids.

[h] UFA, Unsaturated fatty acids.

[i] MUFA, Monounsaturated fatty acids.

[j] PUFA, Polyunsaturated fatty acids.

[k] MCFA, Medium chain fatty acids.

[l] LCFA, Long chain fatty acids.

**Table 8. The effect of herbal mixture on fatty acid proportions (g/100 g FA) in the *longissimus dorsi* muscle of lambs with *H. contortus* infection.**

| Fatty acids, g/100 g FA | CN[a] | CI[a] | M1I[a] | M2I[a] | SEM | *P* value |
|---|---|---|---|---|---|---|
| **Saturated** | | | | | | |
| **C8:0** | 0.10 | 0.12 | 0.15 | 0.25 | 0.02 | 0.08 |
| **C10:0** | 0.10 | 0.19 | 0.15 | 0.32 | 0.04 | 0.20 |
| **C12:0** | 0.61 | 0.85 | 0.65 | 0.71 | 0.07 | 0.64 |
| **C13:0** | 0.26 | 0.90 | 0.29 | 0.78 | 0.13 | 0.16 |
| **C14:0** | 1.05 | 1.00 | 0.68 | 0.62 | 0.09 | 0.28 |
| **C15:0** | 0.32 | 0.19 | 0.18 | 0.17 | 0.03 | 0.22 |
| **C16:0** | 17.7[a] | 16.5[ab] | 14.2[ab] | 13.7[b] | 0.58 | 0.03 |
| **C17:0** | 0.87 | 0.54 | 0.46 | 0.43 | 0.08 | 0.13 |
| **C18:0** | 18.1 | 15.5 | 15.5 | 14.4 | 0.76 | 0.36 |
| **Monounsaturated** | | | | | | |
| **C14:1** | 0.19 | 0.25 | 0.12 | 0.13 | 0.03 | 0.31 |
| **C15:1** | 0.98 | 1.01 | 1.05 | 1.58 | 0.11 | 0.17 |
| **C16:1** | 1.01 | 0.97 | 0.85 | 0.79 | 0.05 | 0.41 |
| **C17:1** | 1.05 | 0.89 | 0.93 | 1.37 | 0.08 | 0.16 |
| **C18:1 *trans*-6-8** | 0.45 | 0.38 | 0.40 | 0.44 | 0.05 | 0.95 |
| **C18:1 *trans*-9** | 0.66 | 0.76 | 0.61 | 0.87 | 0.06 | 0.44 |
| **C18:1 *trans*-10** | 0.76 | 0.79 | 0.51 | 0.53 | 0.08 | 0.46 |
| **C18:1 *trans*-11** | 0.62 | 0.62 | 0.66 | 0.47 | 0.07 | 0.81 |
| **C18:1 *cis*-9** | 24.2 | 21.4 | 20.5 | 22.5 | 0.86 | 0.50 |
| **C18:1 *cis*-11** | 1.49 | 1.52 | 1.53 | 1.44 | 0.04 | 0.89 |
| **C18:1 *cis*-12** | 0.16 | 0.15 | 0.19 | 0.17 | 0.03 | 0.97 |
| **C18:1 *cis*-13** | 0.12 | 0.10 | 0.12 | 0.12 | 0.01 | 0.93 |
| **C18:1 *cis*-14** | 0.13 | 0.25 | 0.17 | 0.16 | 0.03 | 0.57 |
| **Polyunsaturated** | | | | | | |
| **C18:2c9c12** | 13.4 | 13.8 | 16.3 | 15.2 | 0.84 | 0.63 |
| **C18:3 *cis*-9 *cis*-12 *cis*-15 (ALA)[b]** | 1.26 | 1.24 | 1.47 | 1.16 | 0.08 | 0.67 |
| **C18:2 *cis*-9 *trans*-11(RA/CLA)[c]** | 0.11 | 0.10 | 0.10 | 0.11 | 0.02 | 0.98 |
| **C18:2 *trans*-10 *cis*-12** | 0.25 | 0.24 | 0.33 | 0.25 | 0.02 | 0.52 |
| **C18:3n6** | 0.16 | 0.15 | 0.09 | 0.16 | 0.02 | 0.36 |
| **C20:2** | 0.43 | 0.36 | 0.40 | 0.69 | 0.05 | 0.08 |
| **C20:3n6** | 3.47 | 3.20 | 4.42 | 5.04 | 0.30 | 0.09 |
| **C20:4n6** | 0.09 | 0.09 | 0.10 | 0.12 | 0.01 | 0.82 |
| **C20:5n3 (EPA)[d]** | 0.59[b] | 0.65[ab] | 1.11[a] | 0.98[ab] | 0.07 | 0.02 |
| **C22:2** | 0.13 | 0.20 | 0.28 | 0.28 | 0.03 | 0.20 |
| **C22:5n3 (DPA)[e]** | 0.68 | 1.62 | 1.52 | 2.17 | 0.25 | 0.18 |
| **C22:6n3 (DHA)[f]** | 0.30 | 0.42 | 0.25 | 0.37 | 0.03 | 0.19 |
| **SFA[g]** | 44.7 | 46.5 | 43.4 | 39.7 | 1.16 | 0.22 |
| **UFA[h]** | 55.2 | 53.5 | 56.6 | 60.3 | 1.16 | 0.22 |
| **MUFA[i]** | 34.4 | 31.4 | 30.3 | 33.7 | 0.86 | 0.30 |
| **PUFA[j]** | 20.8 | 22.1 | 26.4 | 26.6 | 1.14 | 0.18 |
| **n6 FA** | 17.4 | 17.6 | 21.4 | 21.0 | 1.00 | 0.36 |
| **n3 FA** | 2.83 | 3.94 | 4.35 | 4.68 | 0.29 | 0.09 |
| **n6/n3** | 5.99 | 5.01 | 5.12 | 4.59 | 0.33 | 0.52 |
| **MCFA[k]** | 22.1[a] | 21.6[a] | 18.1[b] | 18.5[b] | 0.60 | 0.01 |
| **LCFA[l]** | 77.7[ab] | 78.1[b] | 81.6[a] | 80.9[a] | 0.58 | 0.02 |
| **Desaturation index** | | | | | | |

*(Continued)*

**Table 8.** (Continued)

| Fatty acids, g/100 g FA | CN[a] | CI[a] | M1I[a] | M2I[a] | SEM | *P* value |
|---|---|---|---|---|---|---|
| **DI (16:1/16)** | 0.06 | 0.06 | 0.06 | 0.05 | 0.00 | 1.00 |
| **DI (18:1/18)** | 0.43 | 0.42 | 0.43 | 0.39 | 0.01 | 0.70 |
| **DI (MUFA/SFA)** | 0.44 | 0.40 | 0.41 | 0.46 | 0.01 | 0.26 |
| **DI (20:4 n6/20:3 n6)** | 0.66 | 0.75 | 0.73 | 0.66 | 0.03 | 0.43 |
| **DI (20:4 n6/18:3 n6)** | 0.33[b] | 0.40[ab] | 0.58[a] | 0.38[b] | 0.03 | 0.03 |
| **DI (22:6 n3/22:5 n3)** | 0.45 | 0.43 | 0.21 | 0.20 | 0.05 | 0.14 |
| **Thrombogenic index** | 0.15 | 0.28 | 0.26 | 0.19 | 0.03 | 0.38 |
| **Atherogenicity index** | 0.69 | 1.07 | 0.96 | 0.72 | 0.08 | 0.31 |

Within each row, means with lower case superscripts (a–c) indicate significant differences at *P* < 0.05; SEM, standard error of the mean.

[a] Control non-infected, CN; Control infected, CI; Mix1 infected, M1I; Mix2 infected, M2I.

[b] ALA, [α]-Linolenic acid.

[c] RA/CLA, Rumenic acid/Conjugated linoleic acid.

[d] EPA, Eicosapentaenoic acid.

[e] DPA, Docosapentaenoic acid.

[f] DHA, Docosahexaenoic acid.

[g] SFA, Saturated fatty acids.

[h] UFA, Unsaturated fatty acids.

[i] MUFA, Monounsaturated fatty acids.

[j] PUFA, Polyunsaturated fatty acids.

[k] MCFA, Medium chain fatty acids.

[l] LCFA, Long chain fatty acids.

**Table 9. Effect of herbal mixture on fatty acid proportions (g/100 g FA) in the subcutaneous fat of lambs with *H. contortus* infection.**

| Fatty acids, g/100 g FA | CN[a] | CI[a] | M1I[a] | M2I[a] | SEM | *P* value |
|---|---|---|---|---|---|---|
| **Saturated** | | | | | | |
| **C8:0** | 0.02 | 0.04 | 0.03 | 0.04 | 0.00 | 0.25 |
| **C10:0** | 0.02 | 0.02 | 0.02 | 0.02 | 0.00 | 0.30 |
| **C12:0** | 0.09 | 0.12 | 0.12 | 0.12 | 0.01 | 0.41 |
| **C13:0** | 0.02 | 0.01 | 0.03 | 0.02 | 0.00 | 0.33 |
| **C14:0** | 1.48 | 1.45 | 1.48 | 1.75 | 0.06 | 0.34 |
| **C15:0** | 0.58[b] | 0.50[b] | 0.61[b] | 0.93[a] | 0.05 | 0.01 |
| **C16:0** | 17.8 | 17.7 | 18.0 | 18.9 | 0.25 | 0.38 |
| **C17:0** | 2.19 | 2.26 | 1.97 | 2.14 | 0.06 | 0.48 |
| **C18:0** | 36.9[ab] | 33.6[b] | 39.5[a] | 38.7[ab] | 0.81 | 0.05 |
| **Monounsaturated** | | | | | | |
| **C14:1** | 0.33[b] | 0.22[b] | 0.34[b] | 0.46[a] | 0.03 | <0.01 |
| **C15:1** | 0.36 | 0.36 | 0.42 | 0.49 | 0.03 | 0.30 |
| **C16:1** | 0.70 | 0.85 | 0.56 | 0.54 | 0.05 | 0.09 |
| **C17:1** | 0.52 | 0.58 | 0.36 | 0.42 | 0.03 | 0.08 |
| **C18:1 *trans*-6-8** | 0.45 | 0.44 | 0.30 | 0.33 | 0.03 | 0.27 |
| **C18:1 *trans*-9** | 0.50 | 0.49 | 0.32 | 0.33 | 0.04 | 0.23 |
| **C18:1 *trans*-10** | 2.81 | 3.55 | 0.73 | 0.68 | 0.55 | 0.16 |
| **C18:1 *trans*-11** | 2.02 | 3.82 | 1.90 | 2.13 | 0.32 | 0.15 |
| **C18:1 *cis*-9** | 21.6 | 22.1 | 22.0 | 20.0 | 0.53 | 0.50 |
| **C18:1 *cis*-11** | 1.34 | 1.58 | 1.21 | 1.31 | 0.05 | 0.10 |

*(Continued)*

**Table 9.** (Continued)

| Fatty acids, g/100 g FA | CN[a] | CI[a] | M1I[a] | M2I[a] | SEM | *P* value |
|---|---|---|---|---|---|---|
| C18:1 *cis*-12 | 0.22[b] | 0.22[b] | 0.25[ab] | 0.28[a] | 0.01 | 0.01 |
| C18:1 *cis*-13 | 0.03 | 0.05 | 0.03 | 0.03 | 0.01 | 0.36 |
| C18:1 *cis*-14 | 0.32[b] | 0.30[b] | 0.42[a] | 0.41[a] | 0.02 | <0.01 |
| Polyunsaturated | | | | | | |
| C18:2 *cis*-9 *cis*-12 | 5.66 | 6.08 | 5.27 | 5.27 | 0.26 | 0.73 |
| C18:3 *cis*-9 *cis*-12 *cis*-15 (ALA)[b] | 0.84[ab] | 0.72[b] | 1.02[a] | 1.00[a] | 0.04 | 0.04 |
| C18:2 *cis*-9 *trans*-11(RA/CLA)[c] | 0.26 | 0.28 | 0.28 | 0.28 | 0.01 | 0.94 |
| C18:2 *trans*-10 *cis*-12 | 0.16 | 0.19 | 0.12 | 0.13 | 0.01 | 0.13 |
| C18:3n6 | 0.04 | 0.04 | 0.04 | 0.04 | 0.00 | 1.00 |
| C20:2 | 0.03 | 0.06 | 0.05 | 0.03 | 0.01 | 0.27 |
| C20:3n6 | 0.17 | 0.32 | 0.21 | 0.38 | 0.04 | 0.36 |
| C20:4n6 | 0.03 | 0.05 | 0.03 | 0.04 | 0.00 | 0.17 |
| C20:5n3 (EPA)[d] | 0.04 | 0.06 | 0.05 | 0.07 | 0.01 | 0.66 |
| C22:2 | 0.13 | 0.10 | 0.12 | 0.16 | 0.01 | 0.23 |
| C22:5n3 (DPA)[e] | 0.10 | 0.08 | 0.14 | 0.16 | 0.03 | 0.78 |
| C22:6n3 (DHA)[f] | 0.27 | 0.28 | 0.22 | 0.21 | 0.02 | 0.42 |
| SFA[g] | 59.8 | 56.3 | 62.3 | 63.3 | 0.99 | 0.06 |
| UFA[h] | 40.2 | 43.7 | 37.7 | 36.7 | 0.99 | 0.06 |
| MUFA[i] | 32.4[ab] | 35.4[a] | 30.2[ab] | 28.9[b] | 0.85 | 0.03 |
| PUFA[j] | 7.74 | 8.27 | 7.54 | 7.76 | 0.31 | 0.91 |
| n6 FA | 6.25 | 6.81 | 5.93 | 6.16 | 0.27 | 0.79 |
| n3 FA | 1.25 | 1.15 | 1.42 | 1.43 | 0.06 | 0.29 |
| n6/n3 | 5.00[ab] | 5.93[a] | 4.14[b] | 4.38[ab] | 0.24 | 0.03 |
| MCFA[k] | 21.4 | 21.3 | 21.5 | 23.2 | 0.32 | 0.08 |
| LCFA[l] | 78.6 | 78.7 | 78.4 | 76.8 | 0.31 | 0.07 |
| Desaturation index | | | | | | |
| DI (16:1/16) | 0.04 | 0.05 | 0.03 | 0.03 | 0.00 | 0.07 |
| DI (18:1/18) | 0.63 | 0.60 | 0.64 | 0.66 | 0.01 | 0.07 |
| DI (MUFA/SFA) | 0.35[ab] | 0.39[a] | 0.33[ab] | 0.31[b] | 0.01 | 0.03 |
| DI (20:4 n6/20:3 n6) | 0.86 | 0.86 | 0.84 | 0.83 | 0.01 | 0.77 |
| DI (20:4 n6/18:3 n6) | 0.42 | 0.54 | 0.40 | 0.46 | 0.04 | 0.64 |
| DI (22:6 n3/22:5 n3) | 4.26 | 3.45 | 3.52 | 1.95 | 0.54 | 0.51 |
| Thrombogenic index | 0.03 | 0.02 | 0.02 | 0.03 | 0.00 | 0.84 |
| Atherogenicity index | 0.50 | 0.44 | 0.51 | 0.55 | 0.02 | 0.27 |

Within each row, means with lower case superscripts (a–c) indicate significant differences at *P* < 0.05; SEM, standard error of the mean.

[a] Control non-infected, CN; Control infected, CI; Mix1 infected, M1I; Mix2 infected, M2I.

[b] ALA, [α]-Linolenic acid.

[c] RA/CLA, Rumenic acid/Conjugated linoleic acid.

[d] EPA, Eicosapentaenoic acid.

[e] DPA, Docosapentaenoic acid.

[f] DHA, Docosahexaenoic acid.

[g] SFA, Saturated fatty acids.

[h] UFA, Unsaturated fatty acids.

[i] MUFA, Monounsaturated fatty acids.

[j] PUFA, Polyunsaturated fatty acids.

[k] MCFA, Medium chain fatty acids.

[l] LCFA, Long chain fatty acids.

**Table 10. The effect of herbal mixture treatment on expression of five genes (lipoprotein lipase (LPL), fatty acid synthase (FASN), stearoyl-CoA desaturase (SCD), fatty acid desaturase 1 (FADS1), fatty acid elongase 5 (ELOVL5), relative transcript abundance) in the *m. longissimus dorsi* of lambs with *H. contortus* infection.**

| Item | CN[a] | CI[a] | M1I[a] | M2I[a] | SEM | *P* value |
|---|---|---|---|---|---|---|
| LPL | 0.86[b] | 2.39[a] | 1.14[ab] | 0.72[b] | 0.21 | 0.01 |
| FASN | 1.15[b] | 2.89[ab] | 1.54[ab] | 3.10[a] | 0.30 | 0.03 |
| SCD | 1.64[b] | 6.62[ab] | 10.3[a] | 1.56[b] | 1.37 | 0.04 |
| FADS1 | 3.04[bc] | 8.59[ab] | 11.7[a] | 0.87[c] | 1.28 | <0.01 |
| ELOVL5 | 6.26 | 7.93 | 10.4 | 5.15 | 1.04 | 0.26 |

Within each row, means with lower case superscripts (a–c) indicate significant differences at *P* < 0.05; SEM, standard error of the mean.

[a]Control non-infected, CN; Control infected, CI; Mix1 infected, M1I; Mix2 infected, M2I.

**Table 11. Lipid peroxidation in serum and oxidative stability of meat in lambs with *H. contortus* infection.**

| Parameter | Day | Dietary treatment group[a] | | | | SEM | *P* value | | |
|---|---|---|---|---|---|---|---|---|---|
| | | CN | CI | M1I | M2I | | G[b] | Time | G × Time |
| Serum TBARS [c] | 22 | 0.24 | 0.24 | 0.19 | 0.27 | 0.013 | 0.099 | <0.001 | 0.059 |
| (μmol/l) | 37 | 0.28 | 0.26 | 0.35 | 0.35 | 0.016 | | | |
| | 51 | 0.30 | 0.36 | 0.31 | 0.33 | 0.016 | | | |
| | 70 | 0.22[a] | 0.33[b] | 0.30[ab] | 0.28[ab] | 0.014 | | | |
| Muscle TBARS | 0 | 0.45 | 0.53 | 0.48 | 0.56 | 0.018 | 0.037 | <0.001 | 0.770 |
| (mg MDA [d]/kg) | 1 | 0.51 | 0.54 | 0.52 | 0.58 | 0.019 | | | |
| | 7 | 0.64[b] | 0.83[a] | 0.63[b] | 0.77[ab] | 0.043 | | | |

Within each row, means with lower case superscripts (a–c) indicate significant differences at P < 0.05; SEM, standard error of the mean.

[a]Control non-infected, CN; Control infected, CI; Mix1 infected, M1I; Mix2 infected, M2I.

[b]G, Group.

[c]TBARS, Thiobarbituric acid reactive substances.

[d]MDA, Malondialdehyde.

33% compared to the free-parasites lambs [42]. Thus, parasite control in ewes can improve production efficiency and may decrease the adverse environmental impacts of sheep production systems. In the present study, methane production was not affected by parasitism. *Archaea* plays a crucial role in methanogenesis, but although the *Archaea* population *in vitro* was slightly diminished, it did not affect methane production. No differences were found both *in vitro* and *in vivo* as the effect of Mix1 or Mix2, could be due to the relatively low content of the anti-methanogenic compounds in the herbal mixtures [43,44,45]. The methane production which showed no differences both in *in vitro* and in *vivo* by Mix1 or Mix2 confirmed the results of the previous study, which presented the interaction of *S. officinalis* basic components and phytochemical compounds causing the reduced antimethanogenic activity due to lower availability of substances for microorganisms [46]. The reduction of the *Archaea* population was not noted *in vivo*, suggesting a lower dose of the herbal mixtures or adaptation of the *Archaea* [47]. Total bacteria and *B. proteoclastus* in the M2I group in *in vivo* study increased. This indicates low concentrations of PSM may stimulate some bacterial populations, while high concentrations of PSM are inhibitory to ruminal microbial populations [48,49]. *Holotricha* population of the CI group was higher compared to the CN group. It may be due to higher susceptibility of *Entodinia* to *H. contortus* infection. *H. contortus* infection alters microbial community composition and diversity, which facilitates the parasite survival and reproduction [50]. Variations in ruminal microbiota composition

response and adaptation to anti-methanogenic compounds, fermentation kinetics, and diet composition are among the major factors contributing to the inconsistent efficacy [51]. The concentrations of total VFA increased in the groups supplemented with herbal mixtures *in vitro*, compared to CN and CI. Observed changes were associated with the increased *in vitro* digestibility in the herbal mixture groups. These results indicated that herbal mixtures perhaps affected the ruminal cellulolytic bacterial activity to increased digestibility (*R. albus*, *R. flavefaciens*, and *F. succinogenes*). Lower concentrations of PSM sometimes may be stimulatory to certain bacterial populations increasing digestibility of feeds. However, significant effects of herbal mixtures on pH, ammonia N and VFA have not been observed *in vivo*, neither in this nor other studies [52], perhaps due to the use of a lower dose of herbal mixture allowing metabolic redundancy of the ruminal ecosystem [49].

Results of *in vitro* FA analyses showed that the infection of *H. contortus* and herbal mixes can modulate the ruminal FA proportion. The infection increased the C18:0 proportion in all infected groups. We hypothesized that the infection increased ruminal microbial lipase activity, the main factor for ruminal BH process [53]. On the other hand, the oxidative stress caused by parasitic infection can stimulate the rumen metabolism of the lambs to fight against the pathogens [54] and hence, the rumen microbial population increased leading to more effective BH process. The decreased effectiveness of BH might be the effect of the antimicrobial properties of PSM against biohydrogenating bacteria [13].

The rumen FA proportion measured in the rumen of lambs did not reflect the results obtained in the *in vitro* experiment. The C14:1 and C17:1 proportion of M2I group slightly increased compared to the CN and CI. The C15:0, C17:0 and total MCFA proportion also increased compared to the CI group. Rumen microbes synthesize odd-chain saturated FA by different pathways, which remove the α-carbon through the conversion of end products of *de novo* lipogenesis (C16:0 and C18:0) to a hydroxyl FA, subsequently by decarboxylation to produce C15:0 and C17:0, respectively [55], or elongation of propionate carbon chain [56]. After absorption, FA proportions were modulated and a numerically higher UFA and lower SFA proportions were found in the blood (Table 6) and liver (Table 7). The PUFA and MUFA proportions in serum were higher than in the rumen, which occurs due to desaturation of FA after absorption from the gastrointestinal tract. Previous studies also showed higher proportion of UFA compared to SFA in ruminants' blood [35], however rumen fluid was characterized with a higher content of SFA [13]. The final values of plasma FA proportions are dependent on the dietary FA source, de novo FA synthesis in tissues, and bacterial synthesis of FA including FA biohydrogenation in the rumen [57,58].

The MUFA proportion in the serum of infected Mix1 group was lower compared to the CN group. The reduced PUFA proportion of infected Mix2 was caused by lower linoleic acid (LA; C18:2n6) content in serum. Moreover, the C16:0; C16:1, C18:1 *cis*-11, conjugated linoleic acid (CLA; C18:2 *cis*-9 *trans*-11) and C20:4n6 proportions in the liver were reduced, while C18:0 and linolenic acid (ALA; C18:3n3) proportions were improved in the herbal mixtures groups. But, no major effect of infection associated with FA proportion was observed in serum and liver of the CN and CI, which were fed a similar type of diet. Therefore, it seems that the bioactive compounds in both herbal mixtures affected the enzymatic lipolysis process, leading to modulation of FA proportions [59]. The C18:3 *cis*-9, *cis*-12, *cis*-15 can be converted to C20:4*n*-6 in the liver by desaturases and elongases, however in the present, study we noticed a lower proportion on C20:4*n*-6 in the liver of lambs fed herbal mixtures, which may suggest the other possible mode of action. In the liver of lambs, the positive effect of M2I was obtained on C18:3 *cis*-9, *cis*-12, *cis*-15, n3 FA, and n6/n3 ration. On the other hand, herbal mixtures both M1I and M12 groups were able to decrease MCFA and increase LCFA, which are also considered favorable within lipid metabolism.

Several studies indicated that diets strongly affected the deposition of intramuscular fat and the proportion of SFA and PUFA [60], as well as the activity of enzymes involved in fatty acids synthesis such as $\Delta$-9 desaturase (converts SFA into *cis*-9 MUFA), elongase (converts C16:0 into C18:0) and $\Delta$-4, $\Delta$-5 and $\Delta$-6 desaturase (convert C18 PUFA into C20-C22 PUFA) [61–64]. A lower biosynthesis of MUFA in the subcutaneous fat of infected Mix2 group was supported by a lower LPL in the infected Mix2 group and a higher SCD activity in the Mix1 group. The SCD is responsible for biosynthesis of *cis*-9, trans-11 CLA from trans-vaccenic acid (C18:1 *trans*-11 CLA) [65]. Therefore, lower LPL activity suggests that biosynthesis of MUFA by the insertion of a double bond between carbon C9 and C10 of SFA, such as stearic acid (C18:0) into oleic acid (C18:1 *cis*-9), is low. In addition, preferential oxidation of FA or competition for desaturation and elongation enzymes by ALA and LA could affect conversion of ALA into a product of metabolites [64]. Moreover, catalytic process for *cis* double bonds into hydrocarbon chains for biosynthesis of UFA increases the n-3 long-chain PUFA, i.e. C20:5 n-3 [66]. Therefore, the C20:5 n-3 was higher in the M1I supported by the FADS1 abundance in muscle, but was lower in the M2I group. Although FADS1 gene expressions in the M2I group decreased, it seems that there is a different mode of action between herbal mixtures groups. Therefore, the results of the present study and those of other researchers suggest that varying FA levels, phytochemical compounds in ruminant diets and varying degree of unsaturation of dietary FA could affect the expression of these lipogenic genes in different ways.

The effects of GIN parasite on the meat quality in sheep had received little attention [26]. Infections with GIN alter energy metabolism to cope with the extra energy required for tackling infection and decrease the body weight of animals [41], which may in turn change FA metabolism. However, in this study, infection did not generally induce major changes in the FA profiles in the tissues, which may be associated with energy utilization by the animal itself. The infection also did not decrease body weight gain in lambs [17]. It has been recognized that the nematode infection induces the production of reactive oxygen, causing oxidative stress in the hosts [67,68]. The concentration of TBARS in meat in the present study showed a constant increase during storage, which indicated that secondary products of lipid oxidation were accumulated during storage. The addition of Mix1, but not Mix2, to the diet of infected lambs exhibited antioxidant potential resulting in a decrease in lipid oxidation in meat by reducing the TBARS level on day 7 of storage as compared to the infected animals. Mix2 herbal mixture had lower concentrations of phenolic and flavonoids compounds than in the Mix1, which was not effective to affect lipid peroxidation in meat. Herbs or forages containing PSM with antioxidative properties also improved meat quality such as chemical composition, colour and lipid stability [69,70].

## Conclusion

Infection did not elicit major impacts on the ruminal fermentation characteristics and FA profiles in tissues, but it increased TBARS in serum and meat after storage. Herbal mixtures supplementation had no effect on the ruminal fermentation characteristics including the ruminal methane production, but increased total VFA concentrations and DM digestibility *in vitro*. Supplementation of herbal mixtures to the diets of GIN parasite infected-lambs decreased MCFA and increased LCFA in liver and meat, and decreased lipid oxidation in meat due to their inhibitory effects on the ruminal biohydrogenation. From this result and previous results [17], it can be concluded that Mix1 may reduce parasitic burdens as well as improve LCFA proportion and oxidative stability in meat, which may prove win-win situations in ruminant production.

## Supporting information

**S1 File.**
(ZIP)

## Acknowledgments

The authors are grateful to Magda Bryszak, Haihao Huang, Yulianri Rizki Yanza and Pawel Kolodziejski for technical assistance.

## Author Contributions

**Conceptualization:** Zora Váradyová, Marián Várady, Adam Cieslak.

**Data curation:** Zora Váradyová.

**Formal analysis:** Paulina Szulc, Dominika Mravčáková, Klaudia Čobanová, Linggawastu Syahrulawal.

**Funding acquisition:** Marián Várady.

**Project administration:** Marián Várady.

**Writing – original draft:** Paulina Szulc.

**Writing – review & editing:** Malgorzata Szumacher-Strabel, Zora Váradyová, Amlan Kumar Patra, Adam Cieslak.

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
