## [Decision Letter · Decision Letter 0]

24 Jan 2020

PONE-D-19-31838

Ruminal fermentation, microbial population and lipid metabolism in gastrointestinal nematode-infected lambs fed a diet supplemented with herbal mixtures

PLOS ONE

Dear Dr Cieslak

Thank you for submitting your manuscript to PLOS ONE. After careful consideration, we feel that it has merit but does not fully meet PLOS ONE’s publication criteria as it currently stands. Therefore, we invite you to submit a revised version of the manuscript that addresses the points raised during the review process.

Many thanks for submitting your manuscript to PLOS One

Your manuscript was reviewed by three experts in the field and they have suggested some changes be made prior to acceptance

If you could write a detailed response to reviewers that would help. For reviewer 3, most of the comments are minor typographical or grammatical issues, so just the word done for each one will be sufficient

the manuscript will be sent back to the same reviewers upon resubmission

I wish you the best of luck with your revision

We would appreciate receiving your revised manuscript by Mar 09 2020 11:59PM. To enhance the reproducibility of your results, we recommend that if applicable you deposit your laboratory protocols in protocols.io, where a protocol can be assigned its own identifier (DOI) such that it can be cited independently in the future. For instructions see: http://journals.plos.org/plosone/s/submission-guidelines#loc-laboratory-protocols

We look forward to receiving your revised manuscript.

Kind regards,

Simon Russell Clegg, PhD

Academic Editor

PLOS ONE

Journal Requirements:

2. In your Methods, please include details of the monitoring of experimental animals for adverse clinical signs.

Reviewers' comments:

Reviewer's Responses to Questions

**Comments to the Author**

1. Is the manuscript technically sound, and do the data support the conclusions?

Reviewer #1: Yes

Reviewer #2: Partly

Reviewer #3: Yes

2. Has the statistical analysis been performed appropriately and rigorously? 

Reviewer #1: Yes

Reviewer #2: Yes

Reviewer #3: Yes

3. Have the authors made all data underlying the findings in their manuscript fully available?

Reviewer #1: Yes

Reviewer #2: Yes

Reviewer #3: Yes

4. Is the manuscript presented in an intelligible fashion and written in standard English?

Reviewer #1: Yes

Reviewer #2: Yes

Reviewer #3: Yes

5. Review Comments to the Author

Reviewer #1: Very interesting article covering functional food (herbal mix), H contortus infection and possible changes in rumen, liver and meat physiology. Innovative aspects of the work bringing new information in this complex interaction.

Reviewer #2: Comments and Suggestions for Authors

I reviewed the manuscript number: PONE-D-19-31838 “ Ruminal fermentation, microbial population and lipid metabolism in gastrointestinal nematode-infected lambs fed a diet supplemented with herbal mixtures”

Please, following some comments on the different sections, and few detailed comments referring to specific lines.

Introduction: The introduction present polyphenols affects but treatment’s herbal mixtures (Mix1 and Mix2) not show the data of polyphenols.

Line80: Fatty acid (FA), another line not uses FA please checks.

Line112: Please show the methods of herbal extract.

Line120: 9 different herbs mixed, that is difficult to separate the affect of herb.

Line122: Mix1 and Mix2 not combine with table 1 check, do you mean herbal mixed or diets?

Line126: Please add black ground of phenolic acids and flavonoids in the introduction.

Line145: Non-infection (CN), Control diet with infection (CI),- non-infected control group (CN), control diet (CI) please check.

Line169: Please delete --- (L3)

Line169: MHco1?

Line236: Please delete --- (PBS)

Discussion

generally, is not complete and discussion not follow the results.

Line553: Methane production was not influenced both in vitro and in vivo by Mix1 or Mix2--- please discussion why different levels of phenolic acids and flavonoids don’t have a affected.

Line555: Archaea population was not noted in vivo suggesting a lower dose of the herbal mixtures--- that convert to material and method, dose of 9 herbal not have a reference

Line556: “B. proteoclastus in the M2I group in in vivo increased, This indicates low concentrations of PSM “ --- I think Control non-infected (CN) and Control infected (CI) low concentrations of PSM more than M2I, please check

Line559: Holotricha population of the CI group was higher compared to the CN group---How different between CI and CN please explain?

Line562:” Interaction of infection… inconsistent efficacy” not relate with the result.

Line 568: non-infected and infected control groups---please change to CN and CI

Line 570: Please add specific name of ruminal cellulolytic bacterial to increased digestibility (R. albus, R. flavefaciens, F. succinogenes)

Line 576: Please check in Line80” The reduced degree of ruminal fatty acid (FA) saturation affects FA composition in ruminant products such as meat and milk”

Line612: Phenolic acids and flavonoids, decreased MCFA and increased LCFA--- what the effect of phenolic acids and flavonoids? please add

Line658: Please delete reference

Table

Table1: CP of Mixed herb --- Why is very high?

Reviewer #3: This is an interesting, well written paper which offers an interesting insight into the use of herbal treatments for disease with gastrointestinal nematodes.

I have made a few comments, mostly minor. I was left asking why and how quite a bit within the methodology, so maybe some additions in here would prove useful.

Although it looks like a lot of comments, most are minor. I am likely to get this back to re-review I would expect, so please don’t bother with a rebuttal to minor comments if you have done them. Only those where you feel a comment is needed is enough for me.

There are also a number of places where it is pretty much constant acronyms, which make it a bit difficult to follow. Is it possible to trim some of these out maybe? (I understand if not)

Line 24- with the gastrointestinal nematode

Line 24- Parallel in vitro ….

Line 34- I think Archaea should not be capitalised

Line 40- 7 should be written in words

Line 61- Maybe better to remove the from the start of the sentence and start, ‘Gastrointestinal parasitic infections ….’

Line 64, comma between production and resulting

Line 69- remove the

Line 71- you mention feeding of plant secondary metabolites to animals- is this as a treatment to those already infected or as more of a prophylaxis to prevent infection?

Line 75- you mention mixed medicinal herbs- a few examples would be nice

Line 76-81- this is a bit repetitive. Consider rewording?

You mention FA composition in meat and milk- is that a good or bad thing? Particularly with the marbelling of waguu beef for example making it highly prized?

Line 85- maybe remove GIN

Line 85- in the present study

Line 87- you choose the musculus longissimus dorsi muscle- is this the best one to choose or is this as a proxy for other muscles?

Line 92- space between profile and have

Line 9- comma between Sciences and in accordance

Line 102- here you talk about the lambs but it would be nice to have some more details, breed, age, sex, weights etc

Line 122-133- I think your mixes of herbs may be clearer in a table, or bullet pointed in a list?

Line 138- was the sample taken from anywhere specific in the rumen as this may affect endothelial cell associated bacterial collection?

Line 138- 139- 6 should be in words

Line 139- how soon post slaughter were the samples taken? And how?

Line 140- observing may sound better than finding out

Line 143- you mention taking samples from different parts of the rumen, again, where and how?

Line 164- again more detail on the lambs, age and sex

Line 166- was the water sterilised or just tap water?

Line 169- infected how? And how do you know that they were viable nematodes?

Line 170- remove GIN

Line 173- I think it should read ‘commercial concentrate was composed of….’

Line 176- DM needs defining

Line 177- you mention animals were housed on a sheep farm. in or outdoors, bedding? With other animals? Biosecurity employed etc?

Line 193- Manufacturer for hot air oven- also what is the method number referring to- it means nothing to the reader

Line 194- manufacturer for muffle furnace

Line 202- define ADF

Line 205- gas production was recorded- how? Using what?

Line 208- space between methanogens). For the ….

Line 213- measuring the molar proportion of ….- how was this done? Using what?

Line 221- space between count and in

Line 238- transferred onto a cellulose disk

I have never heard of an ethanol serie- is it meant to be series? Or does it need a manufacturer?

Line 247- To distinguish

Line 255- lyophilised how?

Line 258- fatty acids were identified- how?

Line 266- space between nitrogen and relative

Line 266- abundance of genes- which genes? How chosen?

Line 268- guessing this is tripure reagent but could be wrong?

Line 269- RNA extraction was performed following manufacturers instructions- which manufacturer?

Manufacturers for:

chloroform (Line 269)

Isopropanol (Line 271)

Ethanol (Line 273)

Thermoblack (Line 274)

DEPC treated water (Line 274)

Line 284- you mention RT_ PCR – a bit more discussion about the genes amplified here would be useful

Line 290- what concentration were primers at?

Line 320- in vitro in italics

Line 322- in vivo in italics

Line 332- a few figures to show the pH decrease may be useful

Line 334- reword to ‘compared to either….’

I thought it a little odd that there was no assessment of the levels of parasite after feeding with the different diets. Is it possible to add that in?

Lines 354-357- as these are the first mention of the different bacterial species, it would be good to have them in full Line 386- ‘…with control diet, as well as ….

Line 389- Holotricha in italics I think

Line 403- by contrast or on the contrary

Line 544- space between nutritional and demand

Line 552- comma after diminished

Line 552- what is the significance, if any of the alteration of Archaea levels?

Line 555- comma after in vivo

Line 556- again, what is the importance of the increase in total bacteria and B. proteoclastus ?

Line 558- population to populations

Line 559- Holotricha in italics I think

Line 562- treatment groups

Line 568- Comma after in vitro

Line 571- sometimes be may sound better

Line 573- comma after in vivo

Line 573- study to studies

Line 574- the use of a lower dose of ….

Line 57—modulate the ruminal FA proportion.

Line 587- space between FA and by

Line 588- remove the α-carbon

Line 594- A previous study, or previous studies

Line 596- with a higher content

Line 606- comma after process

Line 607- space between 15 and can

Line 609- noticed a lower proportion

Line 609- which may suggest …. Sounds better than what can suggest

Line 611-613- this reads a bit unclear- consider rewording

Line 616- space between Δ9 desaturase

Line 621- space between C18:1 trans

Line 635- remove up

Line 636- decrease the body weight of animals

Line 638- attributed to less extent of changes in the energy- reads unclearly- please reword

Line 639- the infection also did not decrease ….

Line 640- nematode infection

Line 658- previous results

6. PLOS authors have the option to publish the peer review history of their article (what does this mean?). If published, this will include your full peer review and any attached files.

Reviewer #1: No

Reviewer #3: No

---

## [Author Response · Author response to Decision Letter 0]

11 Feb 2020

Simon Russell Clegg, PhD

Academic Editor

PLOS ONE

Manuscript ID: PONE-D-19-31838 Ruminal fermentation, microbial population and lipid metabolism in gastrointestinal nematode-infected lambs fed a diet supplemented with herbal mixtures

PLOS ONE

Dear Editor

Dear Reviewers

Thank you very much for evaluating our manuscript and for all valuable suggestions. We follow your instruction and we marked our changes as follows:

- yellow marks words that have been changed, added, or moved to a better location

- the bold text was used for the comments of Referees

-AU: stands for our replies and comments

We would like to thank the Referees for all the valuable comments and suggestions that helped us to improve our manuscript. 

We have introduced all changes indicated in the Editor and Reviewers’ comments and concerns.

Reviewer #1: Very interesting article covering functional food (herbal mix), H contortus infection and possible changes in rumen, liver and meat physiology. Innovative aspects of the work bringing new information in this complex interaction.

We would like to thank very much for positive evaluation. 

Reviewer #2: Comments and Suggestions for Authors

I reviewed the manuscript number: PONE-D-19-31838 “ Ruminal fermentation, microbial population and lipid metabolism in gastrointestinal nematode-infected lambs fed a diet supplemented with herbal mixtures”

Please, following some comments on the different sections, and few detailed comments referring to specific lines.

Introduction: The introduction present polyphenols affects but treatment’s herbal mixtures (Mix1 and Mix2) not show the data of polyphenols.

AU: Our apologies for the misleading information, the main idea of this sentence was to show that feed contains phytochemical substances such as flavonoids or polyphenols can affect ruminal conditions. The proper words were added in line 78-84 to describe PSM that contains phytochemical substances. 

The phytochemical substances that we observed in our treatments were described in line 132-135 for Mix1 and 139-143 for Mix2; we added also a data of phytochemical substances in Table 1 (line 185). Hopefully, this helps for a better understanding as you expected.

Line80: Fatty acid (FA), another line not uses FA please checks.

AU: Thank you for your correction. We apologize for the mistakes. Based on our understanding the aim was to give the consistency of abbreviation after the sentence “Fatty acid (FA)” and for the next sentences that we used the same word was only used the abbreviation “FA”. We have already changed the “Fatty acid” to “FA” (line 37, 42), “volatile fatty acid” to volatile fatty acid (VFA) (line 33).

Line112: Please show the methods of herbal extract.

AU: we are sorry but it was not herbal extract. Mix1 and Mix2 were a mixtures of dry medicinal herbs obtained from commercial sources.

Line120: 9 different herbs mixed, that is difficult to separate the affect of herb.

AU: Thank you for your comment. The emphasis was placed on the phytochemical substances present in herbs mixed in the highest concentration. The effect can be confirmed based on the amount of the phytochemical substances or the mode of actions through the metabolism pathway changing the content of the observed parameters. We added also a data of phytochemical substances in Table 1 (line 186).

eg. 

Item CI Mix1 Mix2 - 

Phenolic acids - 57.3 22.2

EPA in meat 0.65 1.11 0.98

Line122: Mix1 and Mix2 not combine with table 1 check, do you mean herbal mixed or diets?

AU: Thank you for your correction. Mix1 and Mix 2 are herbal mixtures. 

The abbreviations for dietary treatments In vivo are M1I (diet+supplementation of Mix1 + Infected animal ) and M2I (diet+ supplementation of Mix2 + Infected animal )

and for in vitro Control diet with non-infection (CN), Mix1 diet with non-infection (Mix1N), and Mix2 diet with non-infection (Mix2N), Control diet with infection (CI), Mix1 diet with infection (Mix1I), and Mix2 diet with infection (Mix2I).

Hopefully, we answer your question and clarified the issue.

Line126: Please add black ground of phenolic acids and flavonoids in the introduction.

AU: Thank you for the suggestion. We already added a background about phenolic acids and flavonoids in the introduction (line 78-84). We hope this matches your expectations.

Line145: Non-infection (CN), Control diet with infection (CI),- non-infected control group (CN), control diet (CI) please check.

AU: Thank you for your correction. We have already checked and introduced necessary changes (line 152-155) as follows: Control diet with non-infection (CN), Mix1 diet with non-infection (Mix1N), and Mix2 diet with non-infection (Mix2N), Control diet with infection (CI), Mix1 diet with infection (Mix1I), and Mix2 diet with infection (Mix2I). 

Line169: Please delete --- (L3)

AU: We have already deleted. Thank you. (line 176)

Line169: MHco1?

AU: Our apologies for the unclear information. We have already emphasized using brackets to explain that it was the (strain of GIN H. contortus), which is susceptible to all main classes of anthelmintics. (line 177). The nomenclature comes from the system used within Moredun Research Institute, UK.

We hope the explanation is good enough.

Line236: Please delete --- (PBS)

AU: It was deleted. Thank you for the correction. (line 246)

Discussion

generally, is not complete and discussion not follow the results.

AU: We are very grateful for every suggestion and responses are given to improve our manuscript. We have revised the discussion part and edited, hopefully, it is better now.

Line553: Methane production was not influenced both in vitro and in vivo by Mix1 or Mix2--- please discussion why different levels of phenolic acids and flavonoids don’t have a affected.

AU: Thank you for this suggestion. We have already added information about the reason why the bioactive compounds did not affect methane production and we have written as follows: ”Archaea plays a crucial role in methanogenesis, but although the archaea population in vitro was slightly diminished, it did not affect methane production. Not found differences both in vitro and in vivo as the effect of Mix1 or Mix2, could be due to the relatively low content of the anti-methanogenic compounds in the herbal mixtures [43,44,45]. The methane production which showed no differences both in in vitro and in vivo by Mix1 or Mix2 confirmed the results of the previous study which presented the interaction of S. officinalis basic components and phytochemical compounds causing the reduced antimethanogenic activity due to lower availability of substances for microorganisms [46].” (line 569-577)

Line555: Archaea population was not noted in vivo suggesting a lower dose of the herbal mixtures--- that convert to material and method, dose of 9 herbal not have a reference

AU: Thank you for this comment, we have already provided the doses of herbal mixtures in material and methods part and in table 1 (line 186). 

Line556: “B. proteoclastus in the M2I group in in vivo increased, This indicates low concentrations of PSM “ --- I think Control non-infected (CN) and Control infected (CI) low concentrations of PSM more than M2I, please check

AU: Thank you for your specific comment. We have already checked it and it was correct. In in vivo study B. proteoclastus in the M2I group was the highest among all treatments. (line 579)

Line559: Holotricha population of the CI group was higher compared to the CN group---How different between CI and CN please explain?

AU: Holotricha population of the CI group was higher compared to the CN group. We can only speculate that it may be due to higher susceptibility of Entodinia to H. contortus infection. It is known that this parasite alters microbial community composition and diversity, which facilitates the parasite survival and reproduction. It may be the same with Holotricha. However, the mechanism of action is unknown. The above information was mentioned in Lines 581-585

Line562:” Interaction of infection… inconsistent efficacy” not relate with the result.

AU: We apologize for this misinterpretation. Yes, it was not related to our result and it was deleted.

Line 568: non-infected and infected control groups---please change to CN and CI

AU: Thank you for your correction. We did changes already (line 588)

Line 570: Please add specific name of ruminal cellulolytic bacterial to increased digestibility (R. albus, R. flavefaciens, F. succinogenes)

AU: Thank you for your suggestions. We added the name of cellulolytic bacteria as you advised (lines 590-591)

Line 576: Please check in Line80” The reduced degree of ruminal fatty acid (FA) saturation affects FA composition in ruminant products such as meat and milk” 

AU: We have already checked it. Based on your doubts we removed the word ‘reduced’ from the sentence. We would like to highlight that fatty acid present in the rumen and degree of their saturation determine the fatty acid composition in milk and meat. We hope that the explanation is clear enough. Lines 84-85.

Line612: Phenolic acids and flavonoids, decreased MCFA and increased LCFA--- what the effect of phenolic acids and flavonoids? please add

AU: Decreased MCFA and increased LCFA may suggest positive effect of used herbal mixture on lipid metabolism in the liver. However, the precise mechanisms by which flavonoids and phenolic acids exert these actions are not yet fully established, although accumulated data indicated the ability of interaction with lipid metabolism. We have introduced the explanation in Lines 631-633.

Line658: Please delete reference

AU: We deleted reference, thank you.

Table

Table1: CP of Mixed herb --- Why is very high?

AU: CP of Mixed herbs is comparable with quality standards of Hay 1-Hay2

CP of Hay quality standards (g/kg DM)

Prime >190

Hay 1 170-190

Hay 2 140-160

Hay 3 110-130

Hay 4 80-100

Hay 5 < 80

Reviewer #3: This is an interesting, well written paper which offers an interesting insight into the use of herbal treatments for disease with gastrointestinal nematodes.

I have made a few comments, mostly minor. I was left asking why and how quite a bit within the methodology, so maybe some additions in here would prove useful.

Although it looks like a lot of comments, most are minor. I am likely to get this back to re-review I would expect, so please don’t bother with a rebuttal to minor comments if you have done them. Only those where you feel a comment is needed is enough for me.

There are also a number of places where it is pretty much constant acronyms, which make it a bit difficult to follow. Is it possible to trim some of these out maybe? (I understand if not)

AU: We are very grateful for every suggestion and comment we received because it will help to improve our manuscript. We hope that the improvements that we have done can be acceptable.

Line 24- with the gastrointestinal nematode

AU: Thank you for your correction. We fixed it already (line 24)

Line 24- Parallel in vitro ….

AU: Thank you for your correction. We fixed it already (line 24)

Line 34- I think Archaea should not be capitalized

AU: Thank you for your correction. We followed your suggestion (line 34)

Line 40- 7 should be written in words

AU: Thank you for your correction. We have written in words (line 40)

Line 61- Maybe better to remove the from the start of the sentence and start, ‘Gastrointestinal parasitic infections ….’

AU: Thank you for your correction that we followed. (line 61)

Line 64, comma between production and resulting

AU: Thank you for your correction that we followed. (line 64)

Line 69- remove the

AU: Thank you for your correction that we followed. (line 69)

Line 71- you mention feeding of plant secondary metabolites to animals- is this as a treatment to those already infected or as more of a prophylaxis to prevent infection?

AU: This feeding of plant secondary metabolites is dedicated to the animals that are already infected. (lines 71-73)

Line 75- you mention mixed medicinal herbs- a few examples would be nice

AU: Many thanks for your suggestion. We already added some examples which have an effect on reducing the burdens or GIN (line 75-77)

Line 76-81- this is a bit repetitive. Consider rewording?

AU: Thank you for your concern. Yes, we noticed that this was a bit repetitive. We have already rebuild the paragraph (line 79-84)

You mention FA composition in meat and milk- is that a good or bad thing? Particularly with the marbelling of waguu beef for example making it highly prized?

AU: Yes, we agree with the Reviewer but our experiment was done on infected sheep and was discussed taking under consideration the experimental factors used.

Line 85- maybe remove GIN

AU: We have already removed it (line 90)

Line 85- in the present study

AU: Thank you for the correction. We already added “the”. (line 90)

Line 87- you choose the musculus longissimus dorsi muscle- is this the best one to choose or is this as a proxy for other muscles?

AU: We have chosen the musculus longissimus dorsi muscle because intramuscular fat of this part is considered important for evaluating yield and quality of meat. For example, marbling can be defined by the ratio of fat content and muscle mass in this part. Marbling is also associated with physical quality such as tenderness, juice, taste and chemical quality including FA contents which several types are known to have health benefits for humans. On the other hand, it is considered a good part to investigate the protein synthesis increasing the muscle mass. So based on our knowledge, this part is the best one to be chosen (line 92)

Line 92- space between profile and have

AU: Thank you for your correction. We have already fixed it (line 97).

Line 99- comma between Sciences and in accordance

AU: Thank you for your correction. We have already fixed it (line 104)

Line 102- here you talk about the lambs but it would be nice to have some more details, breed, age, sex, weights etc

AU: Thank you for your suggestion. We have already followed your advice (line 107-108)

Line 122-133- I think your mixes of herbs may be clearer in a table, or bullet pointed in a list?

AU: We have added the necessary information in the text

Line 138- was the sample taken from anywhere specific in the rumen as this may affect endothelial cell associated bacterial collection?

AU: Thank you for your question. We have already added that ”The ruminal content was collected from the top, bottom and middle of the rumen of each lamb separately” (line 144-145)

Line 138- 139- 6 should be in words

AU: Thank you. We have already followed you advice (line 146-147)

Line 139- how soon post slaughter were the samples taken? And how?

AU: Thank you for the correction. We added ”Immediately transported to the laboratory in a 39°C preheated water bath” (line 157). Hopefully, this answers your question.

Line 140- observing may sound better than finding out

AU: Thank you. We have already followed you advice (line147)

Line 143- you mention taking samples from different parts of the rumen, again, where and how?

AU: Thank you for the correction. As we already added that rumen digesta was taken from different parts (top, bottom and middle) (line 150)

Line 164- again more detail on the lambs, age and sex

AU: Thank you for the suggestion, we have already added more details of the lambs “Twenty-four Improved Valachian female lambs with an initial mean body weight of 11.7 ± 1.23 kg and 3-4 months of age .”(line 171-173)

Line 166- was the water sterilised or just tap water?

AU: It was drinking tap water Line 174 

Line 169- infected how? And how do you know that they were viable nematodes?

AU: Thank you for your question. As it was already written in the manuscript, infection was through the mouth (orally) with third-stage larvae and was confirmed by our previous experiment that affect the fecal egg counts and number of worms in the animal, which proved the nematodes were viable. (line 177-179)

Line 170- remove GIN

AU: Thank you. We have removed it as you suggested. (line 177)

Line 173- I think it should read ‘commercial concentrate was composed of….’

AU: Thank you, we have fixed it according to your advice (line181)

Line 176- DM needs defining

AU: Thank you, we have fixed it according to your advice (line184)

Line 177- you mention animals were housed on a sheep farm. in or outdoors, bedding? With other animals? Biosecurity employed etc?

AU: The lambs were housed in common stalls on a sheep farm without other animals and with biosecurity employed.

Line 193- Manufacturer for hot air oven- also what is the method number referring to- it means nothing to the reader

AU: Thank you for the suggestion. We have deleted it. Hopefully, it is an appropriate option (line 201)

Line 194- manufacturer for muffle furnace

AU: Thank you, we have added manufacturer for muffle furnace (line202)

Line 202- define ADF

AU: ADF, acid-detergent fiber was explained for the first time in line 206

Line 205- gas production was recorded- how? Using what?

AU: The volume of accumulated gas released from the batch culture was determined from the recorded pressure or the volume of gas produced after 24 h of fermentation using a mechanical manometer fitted to a transducer (Premagas, Stará Turá, Slovak Republic). 

Analysis of gas production was carried out by gas chromatography using a PerkinElmer Clarus 500 gas chromatograph (Perkin Elmer, Inc., Shelton, CT, USA). Lines 213-217

Line 208- space between methanogens). For the ….

AU: Thank you, we have fixed it (line 220)

Line 213- measuring the molar proportion of ….- how was this done? Using what?

AU: Thank you for the correction. We have already added the information that methane production was calculated measuring the molar proportion of VFA in the rumen as follow: 57.5 mol glucose = 65 mol acetate + 20 mol propionate + 15 mol butyrate + 60 mol CO2 + 35 mol CH4 + 25 mol H2O based on Wolin’s equation; Wolin MJ. A Theoretical Rumen Fermentation Balance. J Dairy Sci. 1960;43: 1452–1459. doi:10.3168/jds.S0022-0302(60)90348-9 (line 222-224). Hopefully, we answered your question. Lines 224-227

Line 221- space between count and in

AU: Thank you, we have fixed it (line 234)

Line 238- transferred onto a cellulose disk

AU: Thank you, we have fixed it (line 251)

I have never heard of an ethanol serie- is it meant to be series? Or does it need a manufacturer?

AU: We apologize for the mistakes. It was dehydration in an ethanol concentration level at different levels (500, 800, and 900 ml/L) for 3 min (line 252)

Line 247- To distinguish

AU: Thank you, we have fixed it (line 260)

Line 255- lyophilised how?

AU: Thank you for the response. We have added “lyophilized by freezing, vacuuming and drying the samples (Epsilon 2-10D LSCplus, CHRIST, Germany)” (line 268-269)

Line 258- fatty acids were identified- how?

AU: FA were identified and quantified based on peaks and retention times by comparing FA sample target with appropriate fatty acids methyl ester (FAME) standards (37 FAME Mix, Sigma-Aldrich) and the concentrations of CLAs were determined using a CLA standard (a mixture of cis 9, trans 11 and trans 10, cis 12-octadecadienoic acid methyl esters; Sigma-Aldrich) using a Galaxie Work Station 10.1 (Varian, CA). We hope this gives a better explanation. (line 272-277).

Line 266- space between nitrogen and relative

AU: Thank you, we have fixed it (line 281)

Line 266- abundance of genes- which genes? How chosen?

AU: We apologize for unclear information. We changed to “Relative transcript abundances of five lipogenic genes such as lipoprotein lipase (LPL), fatty acid synthase (FASN), stearoyl-CoA desaturase (SCD), fatty acid desaturase 1 (FADS1), fatty acid elongase 5 (ELOVL5) were measured by real-time PCR method as described previously [10].” Hopefully, this phrase will be proper. (line 281-284)

Line 268- guessing this is tripure reagent but could be wrong?

AU: Thank you for the correction. We fix the words to TriPure reagent (line 284)

Line 269- RNA extraction was performed following manufacturers instructions- which manufacturer?

Manufacturers for:

AU: Chloroform (Line 287) (Sigma Aldrich, Hamburg, Germany)

Isopropanol (Line 289) (Sigma Aldrich, Hamburg, Germany)

Ethanol (Line 291) (POCH, Gliwice, Poland)

Thermoblock (Line 292) (Eppendorf, Hamburg, Germany)

DEPC treated water (Line 293) (Invitrogen, Carlsbad, USA)

TriPure reagent (Line 284) (Roche Diagnostics, Mannheim, Germany).

Line 284- you mention RT_ PCR – a bit more discussion about the genes amplified here would be useful

AU: Thank you for your suggestion. We have introduced necessary explanations (line 304-311)

Line 290- what concentration were primers at?

AU: The concentration was 2 µl of primers mix which was already written in (line 308).

Line 320- in vitro in italics

AU: Thank you for the correction, we have fixed it (line 338)

Line 322- in vivo in italics

AU: Thank you for the correction, we have fixed it (line 340)

Line 332- a few figures to show the pH decrease may be useful

AU: Thank you for the suggestion but we would like to ask the Reviewer to agree for not introducing new figures. The manuscript is already very long. We hope that existing data are clear enough. Please accept our request.

Line 334- reword to ‘compared to either….’

AU: Thank you for the correction, we have fixed the words to “compared to either” (line 352)

I thought it a little odd that there was no assessment of the levels of parasite after feeding with the different diets. Is it possible to add that in?

AU: We would be grateful for your agreement for not adding this information, especially that it was written in Mravčáková et al. (2019) that the anthelmintic potential of herbal mixtures was not sufficient for the primary elimination of parasites, but herbal treatment probably may affect the host over a longer term, reducing the parasitic infection in the host.

Lines 354-357- as these are the first mention of the different bacterial species, it would be good to have them in full Line 386- ‘…with control diet, as well as ….

AU: Thank you for the suggestion. We have already rebuilt the phrase to “The bacteria population (B. fibrisolvens, , R. albus and F. succinogenes) of the infected lambs fed with control diet as well as infected lambs treated with Mix1 and Mix2 diets increased (P < 0.01); however other bacterial populations did not differ among the treatment groups except B. proteoclasticus, which had higher relative abundance in the infected M2I group (P < 0.01)” (line 403-407)

Line 389- Holotricha in italics I think

AU: Thank you for the correction. We have changed to italic (line 407)

Line 403- by contrast or on the contrary

AU: Thank you for suggesting the proper word, we have fixed to “by contrast” (line 421)

Line 544- space between nutritional and demand

AU: Thank you for the correction, we have fixed it (line 561)

Line 552- comma after diminished

AU: Thank you for the correction, we have added comma after diminished (line 568)

Line 552- what is the significance, if any of the alteration of Archaea levels?

AU: Thank you for your question. We rebuild the phrases to follow your suggestion:” Archaea plays a crucial role in methanogenesis, but although the archaea population in vitro was slightly diminished, it did not affect methane production. Not found differences both in vitro and in vivo as the effect of Mix1 or Mix2, could be due to the relatively low content of the anti-methanogenic compounds in the herbal mixtures [43,44,45]. The methane production which showed no differences both in in vitro and in vivo by Mix1 or Mix2 confirmed the results of the previous study which presented the interaction of S. officinalis basic components and phytochemical compounds causing the reduced antimethanogenic activity due to lower availability of substances for microorganisms [46].” (line 569-577)

Line 555- comma after in vivo

AU: Thank you for the correction, we have added comma after in vivo (line 577)

Line 556- again, what is the importance of the increase in total bacteria and B. proteoclastus ?

AU: In our opinion we explained the Reviewer doubt in the next sentence – “This indicates low concentrations of PSM may stimulate some bacterial populations, while high concentrations of PSM are inhibitory to ruminal microbial populations [48,49]” Please accept our explanation (lines 579-581).

Line 558- population to populations

AU: Thank you for the correction, we have fixed it (line 580)

Line 559- Holotricha in italics I think

AU: Thank you for the correction, we have changed into italic (line 581)

Line 562- treatment groups

AU: Thank you for your correction, considering a suggestion from another Reviewer and the fact that it was not related to our results we decided to delete it as the most appropriate option. (It was in line 580)

Line 568- Comma after in vitro

AU: Thank you for your correction. We have added coma after in vitro (line 587)

Line 571- sometimes be may sound better

AU: Thank you for your correction. We have changed to “sometimes may be” (line 591)

Line 573- comma after in vivo

AU: Thank you for your correction. We have added comma after in vivo (line 593)

Line 573- study to studies

AU: Thank you for your correction. We have fixed it as you suggested (line 593)

Line 574- the use of a lower dose of ….

AU: Thank you for your correction. We have fixed it as you suggested (line 594)

Line 577—modulate the ruminal FA proportion.

AU: Thank you for your correction. We have added “the” as you suggested (line 597)

Line 587- space between FA and by

AU: Thank you for your correction. We have fixed it as you suggested (line 607)

Line 588- remove the α-carbon

AU: Thank you for your correction. We have added “the” as you suggested (line 608)

Line 594- A previous study, or previous studies

AU: Thank you for your correction. We have changed it considering plural form (line 614)

Line 596- with a higher content

AU: Thank you for your correction. We have fixed it (lines 615-616)

Line 606- comma after process

AU: Thank you for your correction. We have added comma after process (line 626)

Line 607- space between 15 and can

AU: Thank you for your correction. We have fixed it (line 627)

Line 609- noticed a lower proportion

AU: Thank you for your correction. We have added “a” between 15 and can (line 628)

Line 609- which may suggest …. Sounds better than what can suggest

AU: Thank you for your correction. We have fixed it as you suggested (line 629)

Line 611-613- this reads a bit unclear- consider rewording

AU: Thank you for your response. We rebuild the phrases to “In the liver of lambs, the positive effect of M2I was obtained on C18:3 cis-9, cis-12, cis-15, n3 FA, and n6/n3 ration. On the other hand, herbal mixtures both M1I and M12 groups were able to decrease MCFA and increase LCFA, which is also considered favorable within lipid metabolism” (line 630-633). We hope this version is better to understand.

Line 616- space between Δ9 desaturase

AU: Thank you for your correction. We have fixed it as you suggested (line 636)

Line 621- space between C18:1 trans

AU: Thank you for your correction. We have fixed it as you suggested (lines 640-641)

Line 635- remove up

AU: Thank you for your correction. We have removed “up” as you suggested (line 654)

Line 636- decrease the body weight of animals

AU: Thank you for your correction. We have added “the” as you suggested (line 655)

Line 638- attributed to less extent of changes in the energy- reads unclearly- please reword

AU: Thank you for your correction. We have reworded to “However, in this study, infection did not generally induce major changes in the FA profiles in the tissues, which may be associated with energy utilization by the animal itself” (lines 656-656)

 We hope this more clear and better to understand. 

Line 639- the infection also did not decrease ….

AU: Thank you for your correction. We have fixed it as you suggested (line 658)

Line 640- nematode infection

AU: Thank you for your correction. We have fixed it as you suggested (line 659)

Line 658- previous results

AU: Thank you for your correction. We have fixed it as you suggested (line 677)

We would like once more to thank very much the Editor and Reviewers for all the valuable comments and suggestions that helped us to improve our evaluated manuscript.

---

## [Decision Letter · Decision Letter 1]

10 Mar 2020

PONE-D-19-31838R1

Ruminal fermentation, microbial population and lipid metabolism in gastrointestinal nematode-infected lambs fed a diet supplemented with herbal mixtures

PLOS ONE

Dear Dr Cieslak

Thank you for submitting your manuscript to PLOS ONE. After careful consideration, we request that you make a few, very minor modifications to it as recommended by the reviewers prior to recommendation for publication. Therefore, we invite you to submit a revised version of the manuscript that addresses the points raised during the review process.

Many thanks for submitting your revised manuscript to PLOS One

It was reviewed by the same two experts in the field as the last submission, and both have suggested some very minor revisions should be made before it is accepted for publication.

If you could write a brief response to reviewer 2 comments that would be helpful. As reviewer 3comments are so minor, just a simple line saying these have been done will be enough.

I will then read the manuscript, and providing comments are addressed, I will recommend it for publication without the need to re-review

I wish you the best of luck with your revisions

Many thanks

Simon

We would appreciate receiving your revised manuscript by Apr 24 2020 11:59PM. To enhance the reproducibility of your results, we recommend that if applicable you deposit your laboratory protocols in protocols.io, where a protocol can be assigned its own identifier (DOI) such that it can be cited independently in the future. For instructions see: http://journals.plos.org/plosone/s/submission-guidelines#loc-laboratory-protocols

A marked-up copy of your manuscript that highlights changes made to the original version. This file should be uploaded as separate file and labeled 'Revised Manuscript with Track Changes'.An unmarked version of your revised paper without tracked changes. This file should be uploaded as separate file and labeled 'Manuscript'.

We look forward to receiving your revised manuscript.

Kind regards,

Simon Russell Clegg, PhD

Academic Editor

PLOS ONE

Reviewers' comments:

Reviewer's Responses to Questions

**Comments to the Author**

1. If the authors have adequately addressed your comments raised in a previous round of review and you feel that this manuscript is now acceptable for publication, you may indicate that here to bypass the “Comments to the Author” section, enter your conflict of interest statement in the “Confidential to Editor” section, and submit your "Accept" recommendation.

Reviewer #2: All comments have been addressed

Reviewer #3: All comments have been addressed

2. Is the manuscript technically sound, and do the data support the conclusions?

Reviewer #2: Partly

Reviewer #3: Yes

3. Has the statistical analysis been performed appropriately and rigorously? 

Reviewer #2: Yes

Reviewer #3: Yes

4. Have the authors made all data underlying the findings in their manuscript fully available?

Reviewer #2: Yes

Reviewer #3: Yes

5. Is the manuscript presented in an intelligible fashion and written in standard English?

Reviewer #2: Yes

Reviewer #3: Yes

6. Review Comments to the Author

Reviewer #2: Even though, authors have been revised accordingly to my comments, there are minor point need more address.

L82-83: provide detail mechanism how polyphenols inhibit the populations of microbes.

L143-144: what kind of feed was fed to lamb? Detail.

L671: Conclusion is too hard understand, please revise and make to related to hypothesis and objective study.

Reviewer #3: I wish to thank the authors for their detailed review response to the comments which I made on the last manuscript. I know many were very minor, and thank you for addressing them, and the comments to the others which are all acceptable to me (most were asked out of curiosity).

I have made a few very minor comments, which if addressed, I would not expect to see a revised version prior to publication.

Thanks once again, and best wishes

Line 107- remove were obtained from the same farm- as you say that twice

Line 173- after the adaptive period

Line 303- you may wish to say if any of these genes were sequenced as part of the PCR development

Line 571- not found differences doesn’t make sense- maybe no differences were found?

Line 575- comma after study may make this sentence a bit easier to read

Disucssion- I believe that Archaea needs capitalising throughout

7. PLOS authors have the option to publish the peer review history of their article (what does this mean?). If published, this will include your full peer review and any attached files.

Reviewer #3: No

---

## [Author Response · Author response to Decision Letter 1]

11 Mar 2020

Poznan, 11.03.2020 

Simon Russell Clegg, PhD

Academic Editor

PLOS ONE

Manuscript ID: PONE-D-19-31838 Ruminal fermentation, microbial population and lipid metabolism in gastrointestinal nematode-infected lambs fed a diet supplemented with herbal mixtures

PLOS ONE

Dear Editor

Dear Reviewers

Thank you very much for evaluating again our manuscript and minor suggestions. We follow your instruction and we marked our changes as follows:

- yellow marks words that have been changed, added, or moved to a better location

- the bold text was used for the comments of Referees

-AU: stands for our replies and comments

We would like to thank the Referees for all the valuable comments and suggestions that helped us to improve our manuscript. 

We have introduced all changes indicated in the Editor and Reviewers’ comments and concerns.

Reviewer #2: Even though, authors have been revised accordingly to my comments, there are minor point need more address.

L82-83: provide detail mechanism how polyphenols inhibit the populations of microbes.

AU: Thank you for your suggestion. We added the following information L82-85:

Polyphenols inhibit the populations and/or activity of microbes responsible for methanogenesis and biohydrogenation by among others changing the rumen environment (pH value) and through the toxic effect on methanogens, consequently lowering methane emission and biohydrogenation rate of UFA in the rumen.

L143-144: what kind of feed was fed to lamb? Detail.

AU: In our opinion all detailed information regarding lambs feeding are located in L 118-121

Animals were fed a concentrate mixture (500 g dry matter (DM)/d), herbal mixtures (non-commercial mixtures - Mix1 and Mix2; 100 g DM/d) and meadow hay (ad libitum). The concentrate mixture was composed of 700 g/kg of barley, 220 g/kg of soybean meal, 48 g/kg of wheat bran, 5 g/kg of bicarbonate and 27 g/kg of mineral-vitamin premix.

Because of it we would like kindly to ask the Reviewer to keep as it is.

L671: Conclusion is too hard understand, please revise and make to related to hypothesis and objective study.

AU: Thank you for your suggestion. We improved our conclusion, and we hope you will accept it.

Now the conclusion is as follow:

L 674-683

Infection did not elicit major impacts on the ruminal fermentation characteristics and FA profiles in tissues, but it increased TBARS in serum and meat after storage. Herbal mixtures supplementation had no effect on the ruminal fermentation characteristics including the ruminal methane production, but increased total VFA concentrations and DM digestibility in vitro. Supplementation of herbal mixtures to the diets of GIN parasite infected-lambs decreased MCFA and increased LCFA in liver and meat, and decreased lipid oxidation in meat due to their inhibitory effects on the ruminal biohydrogenation. From this result and previous results [17], it can be concluded that Mix1 may reduce parasitic burdens as well as improve LCFA proportion and oxidative stability in meat, which may prove win-win situations in ruminant production.

Reviewer #3: I wish to thank the authors for their detailed review response to the comments which I made on the last manuscript. I know many were very minor, and thank you for addressing them, and the comments to the others which are all acceptable to me (most were asked out of curiosity).

I have made a few very minor comments, which if addressed, I would not expect to see a revised version prior to publication.

Thanks once again, and best wishes

Line 107- remove were obtained from the same farm- as you say that twice

AU: Thank you for your correction. It has been done.

Line 173- after the adaptive period

AU: It was corrected. Thank you.

Line 303- you may wish to say if any of these genes were sequenced as part of the PCR development

AU: We are sorry for misunderstanding. Transcript expression of the 5 genes investigated in this study was carried out with the RT-qPCR method. Any of the sequencing procedure has not been applied.

Line 571- not found differences doesn’t make sense- maybe no differences were found?

AU: It was improved. Thank you.

Line 575- comma after study may make this sentence a bit easier to read

AU: Thank you – you are right. We introduced necessary changes into the following sentence:

The methane production which showed no differences both in in vitro and in vivo by Mix1 or Mix2 confirmed the results of the previous study, which presented the interaction of S. officinalis basic components and phytochemical compounds causing the reduced antimethanogenic activity due to lower availability of substances for microorganisms [46].

Disucssion- I believe that Archaea needs capitalising throughout

AU: It was corrected. Thank you.

We would like once more to thank very much the Editor and Reviewers for all the valuable comments and suggestions that helped us to improve our evaluated manuscript.

---

## [Editor Report · Decision Letter 2]

26 Mar 2020

Ruminal fermentation, microbial population and lipid metabolism in gastrointestinal nematode-infected lambs fed a diet supplemented with herbal mixtures

PONE-D-19-31838R2

Dear Dr. Cieslak

We are pleased to inform you that your manuscript has been judged scientifically suitable for publication and will be formally accepted for publication once it complies with all outstanding technical requirements.

With kind regards,

Simon Russell Clegg, PhD

Academic Editor

PLOS ONE

Additional Editor Comments:

Line 104- change animal to animals

Line 302- put a semi colon before the cycling conditions

Line 579- space after mention of reference 46

Line 631- comma after present study

Line 645- change bound to bond

Line 662- change induce to induces

Many thanks for resubmitting your manuscript to PLOS One and for addressing previous reviewers comments

I have reviewed the manuscript and recommended it for acceptance and publication

It has been a pleasure working with you, and I wish you all the best for your future research

Hope you are well and keeping safe in these difficult times

Best wishes and thanks

Simon

---

## [Editor Report · Acceptance letter]

30 Mar 2020

PONE-D-19-31838R2 

Ruminal fermentation, microbial population and lipid metabolism in gastrointestinal nematode-infected lambs fed a diet supplemented with herbal mixtures 

Dear Dr. Cieslak:

I am pleased to inform you that your manuscript has been deemed suitable for publication in PLOS ONE. Congratulations! Your manuscript is now with our production department. 

With kind regards,

on behalf of

Dr. Simon Russell Clegg 

Academic Editor

PLOS ONE